# Inhibition of HER2 signaling and breast cancer cell growth with a novel antibody targeting HER2 ECD III/IV

Chunchun Liu[1]*, Yulei Ren[1], Xuan Luo[2,3], Peiyun Zhou[1], Xiangjuan Du[1], Qianmin Wang[1], Xue Han[1], Yanhui Xu[1,4,5], Ping Wang[6], Dan Zhao[1,7]*, Huirong Yang[1,4,5]*

1 Fudan University Shanghai Cancer Center, Institutes of Biomedical Sciences, Shanghai Key Laboratory of Radiation Oncology, and Shanghai Key Laboratory of Medical Epigenetics, Shanghai Medical College of Fudan University, Shanghai, China, 2 Key Laboratory of Breast Cancer in Shanghai, Department of Breast Surgery, Fudan University Shanghai Cancer Center, Shanghai Medical College, Fudan University, Shanghai, China, 3 Department of Oncology, Shanghai Medical College, Fudan University, Shanghai, China, 4 The International Co-laboratory of Medical Epigenetics and Metabolism, Ministry of Science and Technology, Shanghai, China, 5 Department of Systems Biology for Medicine, School of Basic Medical Sciences, Shanghai Medical College of Fudan University, Shanghai, China, 6 Molinsights Biotechnologies, Guangdong Medical Valley Life Science Park, Guangzhou, China, 7 School of Life Sciences, Greater Bay Area Institute of Precision Medicine (Guangzhou), Fudan University, Guangzhou, China

☯ These authors contributed equally to this work.
* liuchun880616@yeah.net (CL); danzhao5278@foxmail.com (DZ); yanghr@fudan.edu.cn (HY)

## Abstract

Human epidermal growth factor receptor 2 (HER2), a ligand-independent tyrosine kinase receptor belonging to the EGFR family, serves as a key oncogenic driver in breast, gastric, and several other solid tumors. Although anti-HER2 therapies have substantially improved survival outcomes—particularly in breast and gastric cancers—treatment resistance and cancer recurrence remain major clinical challenges. Thus, developing novel antibodies exhibiting complementary effects to the current anti-HER2 therapies could provide additional therapeutic benefits. Here, we describe two novel HER2-targeting monoclonal antibodies, m66 (murine-derived) and r40 (rabbit-derived), which inhibit breast cancer cell proliferation *in vitro*. Of the two, antibody r40 exhibits stronger suppression of the PI3K/AKT and MAPK signaling pathways compared to m66. Moreover, the addition of r40 to the combination of trastuzumab and pertuzumab leads to a significantly enhanced inhibitory effect. We also determined the cryo-EM structures of the HER2-m66-trastuzumab ternary complex and the HER2-r40-trastuzumab-pertuzumab tetrameric complex, at overall resolutions of 3.2 Å and 3.1 Å, respectively. Structural analyses reveal that m66 recognizes an epitope overlapping with that of pertuzumab, whereas r40 binds within the HER2 ECD III/IV—a region distinct from the binding sites of both trastuzumab and pertuzumab. These findings identify r40 as a promising therapeutic candidate for use in combination with trastuzumab and pertuzumab in the treatment of HER2-positive breast cancer.

**Data availability statement:** The cryo-EM maps have been deposited in the EM Databank under accession codes: EMDB-68255 (HER2-m66-trastuzumab), EMDB-68256 (HER2-r40-trastuzumab-pertuzumab). The atomic coordinates have been deposited in the Protein Data Bank with accession numbers: 22GC (HER2-m66-trastuzumab), 22GE (HER2-r40-trastuzumab-pertuzumab).

**Funding:** This work was supported by grants from the Shanghai Natural Science Foundation of China (21ZR1407900). The funders had played a role in study design, data collection analysis, decision to publish, and preparation of the manuscript.

**Competing interests:** The authors have declared that no competing interests exist.

## Introduction

The human epidermal growth factor receptor (EGFR) family consists of four members: EGFR (ERBB1/HER1), HER2 (ERBB2), HER3 (ERBB3), and HER4 (ERBB4). Each can act as an oncogenic driver, with EGFR and HER2 being particularly strongly associated with tumorigenesis. HER2 is a single-pass type I transmembrane protein composed of four extracellular domains (ECD I-IV), a transmembrane domain, and an intracellular tyrosine kinase domain, arranged from the N- to C-terminus. Unlike other EGFR family members, HER2 is constitutively active and can trigger downstream signaling cascades in the absence of an activating ligand [1,2]. Overexpression of HER2 promotes both ligand-independent homodimerization and ligand-dependent heterodimerization with other family members—such as EGFR/HER2, HER2/HER3, and HER2/HER4—resulting in HER2 phosphorylation, activation of the PI3K/AKT and MAPK signaling pathways, and ultimately stimulation of cell proliferation and growth [1,3].

Current clinical guidelines incorporate a range of anti-HER2 biologics—from monoclonal antibodies to advanced antibody-drug conjugates (ADCs)—for the treatment of cancers with amplified HER2 expression. Trastuzumab (Herceptin®), approved in 1998 for breast cancer and in 2010 for metastatic gastric cancer, is indicated for patients with HER2 overexpression. It binds to ECD IV of HER2, inducing receptor internalization and degradation. These effects inhibit downstream signaling pathways and elicit antibody-dependent cellular cytotoxicity (ADCC), leading to tumor cell elimination through immune effector mechanisms [4–6]. When used in combination with chemotherapy, trastuzumab has demonstrated significant therapeutic benefits in HER2-driven breast malignancies. Nevertheless, resistance to therapy and disease relapse remain common challenges following trastuzumab treatment [7]. Pertuzumab (Perjeta®), the second HER2-targeting monoclonal antibody approved for clinical use, binds to ECD II of HER2 [8,9]. The binding epitopes recognized by trastuzumab and pertuzumab are both located at the dimerization interface of HER2 [10]. Combination therapy with trastuzumab and pertuzumab has been shown to enhance antitumor efficacy in HER2-overexpressing breast cancer [11–13].

Other anti-HER2 agents, including small-molecule inhibitors (lapatinib, neratinib), and ADC drugs (trastuzumab emtansine, trastuzumab deruxtecan) have also been approved for HER2-overexpression breast cancers [14–18]. Furthermore, both trastuzumab and trastuzumab deruxtecan (T-Dxd) are approved for the treatment of HER2-positive gastric cancer [19–21]. Despite these advances, metastatic HER2-driven malignancies often develop resistance to the existing targeted therapies, ultimately leading to disease progression [22].

The demonstrated synergy between trastuzumab and pertuzumab—which target distinct epitopes on HER2—has spurred considerable interest in developing novel antibodies that recognize new epitopes [23]. Studies have shown that combinations of three anti-HER2 antibodies targeting non-overlapping epitopes can achieve superior antitumor efficacy compared to the dual combination of trastuzumab and pertuzumab [24]. Moreover, the addition of a monoclonal antibody targeting HER2 ECD III

to trastuzumab and pertuzumab has been shown to enhance the blockade of HER2 heterodimerization and downstream signaling [25]. Although neither ECD I nor ECD III is directly involved in HER2 dimerization, ECD I plays a key role in forming a binding pocket for the dimerization arm of other EGFR family members; ECD III, located centrally within the extracellular domain, serves as a critical structural bridge between ECD II and ECD IV and interacts directly with ECD I [26–28]. Antibodies targeting ECD I or ECD III may modulate the conformational flexibility of HER2, potentially interfering with its signaling activation cascade [23].

In this study, we identify two novel HER2-targeting antibodies, m66 (murine-derived) and r40 (rabbit-derived). The rabbit monoclonal antibody r40 specifically binds to HER2 ECD III/IV and, similar to trastuzumab, inhibits breast cancer cell proliferation by suppressing the PI3K/AKT and MAPK signaling pathways through reduction of HER2 protein expression. Furthermore, r40 significantly enhances the antitumor efficacy of the trastuzumab-pertuzumab combination, leading to more potent suppression of breast cancer cell growth and HER2-mediated signaling.

## Results

### Discovery of antibodies targeting HER2 ECD

HER2 ECD-specific antibodies were obtained by single-cell VDJ sequencing (scVDJ-seq) from the spleens of mice and rabbits immunized with recombinant HER2 ECD protein (S1A Fig). Selected antibody sequences derived from antigen-reactive B cells were synthesized, expressed *in vitro* and evaluated for binding affinity using pull-down and enzyme-linked immunosorbent assay (ELISA) assays (S1D–S1F Fig). Through this screening process, we identified two murine antibodies (designated m66 and m75) and 38 rabbit antibodies (labeled r1 through r40) that specifically bound to the HER2 ECD.

### Antibody m66 competes with pertuzumab for binding to HER2

To characterize the epitopes of the two murine antibodies, we purified trastuzumab, pertuzumab, m66, and m75 in both His-tagged and non-His-tagged human IgG1 formats (S1B Fig). These variants were generated by including or omitting HRV 3C protease cleavage during purification. The $EC_{50}$ of each antibody was determined by ELISA using immobilized HER2 ECD (S1G Fig). Among the tested agents, r40 exhibited the highest affinity, slightly exceeding that of trastuzumab. Pertuzumab showed intermediate affinity, while m75 demonstrated moderately reduced binding. In contrast, m66 displayed the lowest affinity of all antibodies evaluated (S1G Fig). We then performed a competitive ELISA in which immobilized HER2 was pre-incubated with a saturating concentration (2 ng/µL) of non-His-tagged IgG, followed by assessment of binding using His-tagged IgG. As anticipated, each non-His-tagged IgG effectively inhibited the binding of its His-tagged counterpart (Fig 1A). Furthermore, pre-saturation with m66 or m75 did not impair trastuzumab binding, suggesting that both antibodies bind to epitopes distinct from that of trastuzumab. Interestingly, the binding of m66 was impaired by pertuzumab, whereas pertuzumab binding was not affected by m66, indicating that m66 has lower binding affinity than pertuzumab and may bind an overlapping or adjacent epitope (Fig 1A). Antibody m75 reduced the binding of pertuzumab, and conversely, pertuzumab impaired the binding of m75 (Fig 1A). These findings suggest that m66 and m75 recognize epitopes distinct from that of trastuzumab, yet likely overlap with the binding site of pertuzumab.

To map antibody epitope via flow cytometry, we used the low-HER2-expressing breast cancer cell line MCF7 [2]. This model enabled us to evaluate whether the candidate antibodies maintained HER2-binding capability in the presence of saturating concentrations of trastuzumab and pertuzumab. To determine the saturating concentrations of trastuzumab and pertuzumab on MCF7 cells, we incubated the cells with each antibody across a concentration range of 0–400 nM. After incubation, the cells were stained with an AF647-conjugated goat anti-human secondary antibody and analyzed by flow cytometry. As shown in the saturation curve, the saturating concentrations for both antibodies were determined to be approximately 133.3 nM (Fig 1B).

**Fig 1. Epitope characterization by ELISA and flow cytometry.** A. Competitive ELISA assessing the binding of His-tagged IgGs to immobilized HER2 in the presence or absence of saturating concentrations of non-His-tagged IgGs, detected with an anti-His-HRP antibody. B. Saturation binding curves of trastuzumab and pertuzumab to HER2 on live MCF7 cells. Cells were incubated with antibodies (0-400 nM) for 30 min and detected with an AF647-conjugated goat anti-human secondary antibody via flow cytometry. C. Competitive binding of of AF647-labeled trastuzumab or pertuzumab (0-300 nM) to MCF7 cells pre-saturated with 267 nM of unlabeled trastuzumab and pertuzumab, analyzed by flow cytometry. D. Binding of AF647-conjugated m66 or m75 to MCF7 cells after pre-incubation with 267 nM trastuzumab or pertuzumab. E. Binding of AF647-conjugated rabbit antibodies to MCF7 cells pre-saturated with 267 nM of trastuzumab and pertuzumab, evaluated by flow cytometry. T, trastuzumab; P, pertuzumab. Error bars, SD. *, $P < 0.05$; **, $P < 0.01$; ***, $P < 0.001$; ****, $P < 0.0001$, T-test.

Subsequently, trastuzumab, pertuzumab, m66, and m75 were conjugated with NHS-AF647. As expected, the binding of AF647-conjugated trastuzumab and pertuzumab was abolished in MCF7 cells pre-saturated with 267 nM (twice the saturation concentration) of unlabeled trastuzumab and pertuzumab (Fig 1C). Similarly, the binding of both m66-AF647 and m75-AF647 was blocked exclusively by pre-saturation with unlabeled pertuzumab, but only minimally affected by trastuzumab saturation (Fig 1D). These results further support that m66 and m75 recognize epitopes overlapping with that of pertuzumab, consistent with the ELISA data (Fig 1A). Notably, increased m66-AF647 binding was observed in cells pre-saturated with trastuzumab, suggesting that trastuzumab binding induces conformational changes in HER2 that enhance the accessibility or affinity for m66 (Fig 1D). During antibody purification, we noted that m75 exhibited instability and a tendency to precipitate; therefore, m66 was selected for further characterization.

## Antibody r40 binds to a different epitope from that of trastuzumab and pertuzumab

We initially cloned the full-length HER2 protein and expressed it in Expi293 cells to evaluate antibody binding. However, we observed that untransfected Expi293 cells also exhibited binding to the HER2 antibodies. Hence, we proceeded with subsequent experiments without transfection. Initially, we tested the binding of 39 rabbit antibodies to Expi293 cells. Among these, antibodies r22, r30, r35 showed no binding to cell-surface HER2 (S2A Fig). To characterize rabbit antibody epitopes through competitive binding, we pre-incubated Expi293 cells with both trastuzumab and pertuzumab at the concentration of 10 µg/mL (66.7 nM). The cells were then incubated with individual rabbit antibodies and analyzed by flow cytometry to evaluate simultaneous binding. For the antibodies that did bind, pre-incubation with trastuzumab and pertuzumab did not reduce the mean fluorescence intensity (MFI) compared to the control (cells incubated with the rabbit antibody alone). This indicated that the binding of these rabbit antibodies to HER2 did not compete with the epitopes targeted by trastuzumab and pertuzumab. Based on these results, we concluded that six rabbit antibodies (r3, r4, r5, r29, r36, and r40) might recognize epitopes distinct from those of trastuzumab and pertuzumab (S2B Fig).

The epitope recognition profiles of r3, r4, r5, r29, r36, and r40 were subsequently characterized and compared with those of trastuzumab and pertuzumab using the low-HER2-expressing MCF7 cells (Fig 1E). Pre-saturation of HER2 with trastuzumab and pertuzumab did not impede the subsequent binding of AF647-conjugated r3, r4, r5, r29, or r40, suggesting that their epitopes are distinct from those targeted by these two therapeutic antibodies (Fig 1E). In contrast, r36 exhibited enhanced binding signal on cells pre-saturated with trastuzumab and pertuzumab, suggesting that therapeutic antibody binding induces conformational changes in HER2 that facilitate r36 recognition (Fig 1E).

## Antibody r40 inhibits breast cancer cell proliferation

Next, we evaluated the anti-proliferative activity of six rabbit antibodies (r3, r4, r5, r29, r36, r40) and the mouse antibody m66 in HER2-overexpressing BT-474 breast cancer cells (Fig 2A). Antibody r40 demonstrated concentration-dependent suppression of cell viability, outperforming the potency of both trastuzumab and pertuzumab. The triple combination of trastuzumab, pertuzumab and r40 enhanced inhibitory activity at concentrations below 10 nM compared to the dual combination

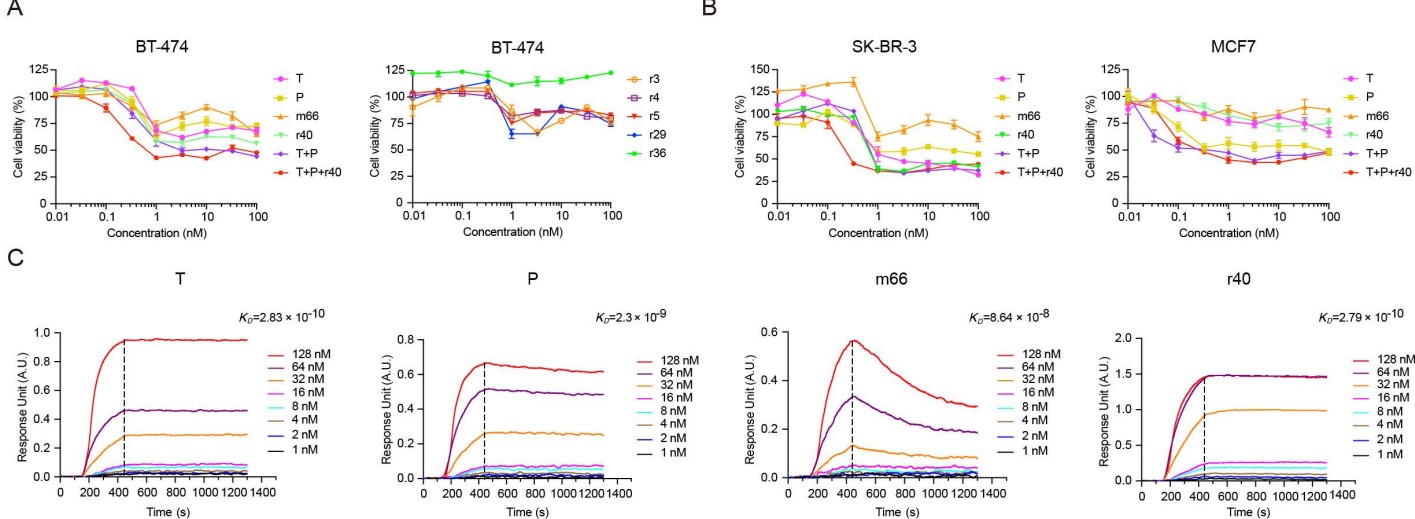

**Fig 2. Functional characterization of m66 and r40.** A. Viability of BT-474 cells following 4 d treatment with a 1:3 serial dilution (0-100 nM) of the indicated antibodies, as assessed by CCK-8 assay. B. Dose-dependent inhibition of SK-BR-3 and MCF7 cell proliferation after 4 d of exposure to antibody dilutions (0-100 nM). Data are presented as mean ± SEM from n = 5 (BT-474 and SK-BR-3) and n = 6 (MCF7) independent experiments. C. SPRi analysis of the binding kinetics between HER2 and trastuzumab, pertuzumab, m66 and r40. Antibodies (0.5 mg/mL) were immobilized on a 3D optical crosslinking biosensor chip. HER2 was injected at concentrations ranging from 1 to 128 nM. Association and dissociation phases were monitored for 285 s and 930 s, respectively, using a PlexArray® HT system.

(trastuzumab and pertuzumab) or single agents. In contrast, the other five rabbit antibodies and m66 exhibited limited effects on cell proliferation (Fig 2A). Based on its superior efficacy, r40 was selected for further analysis, alongside m66.

In two additional breast cancer cell lines—SK-BR-3 (HER2-overexpressing) and MCF7 (low-HER2-expressing)—r40 demonstrated anti-proliferative activity comparable to that of trastuzumab and pertuzumab (Fig 2B). Furthermore, the triple combination showed enhanced or similar efficacy compared to the dual combination (Fig 2B).

The EC$_{50}$ values of the antibodies and the combinations were determined using CCK-8 proliferation assays followed by non-linear regression analysis (log[inhibitor] vs. normalized response – variable slope) in GraphPad Prism (S2C Fig). In both BT-474 and SK-BR-3 cells, the triple combination demonstrated superior efficacy, surpassing that of all single-agent and dual treatments (S2C Fig).

The binding kinetics of trastuzumab, pertuzumab, m66 and r40 to HER2 were evaluated by Surface Plasmon Resonance Imaging (SPRi). Notably, r40 exhibited high affinity ($K_D$ = 0.279 nM), comparable to that of trastuzumab ($K_D$ = 0.283 nM). In contrast, m66 showed the modest binding affinity ($K_D$ = 86.4 nM), which was considerably lower than that of both r40 and the therapeutic antibodies trastuzumab and pertuzumab ($K_D$ = 2.3 nM) (Fig 2C). These results indicate that r40 possesses excellent HER2-binding properties, whereas m66 demonstrates relatively weak binding under the tested conditions.

## Antibody r40 and m66 could bind and detect cell-surface HER2 of breast cancer cells

We next evaluated the binding capacity of m66 and r40 to the endogenous HER2 protein in breast cancer cells using flow cytometry. The m66 antibody, which incorporates variable regions on a human IgG1 backbone, can be detected using an anti-human secondary antibody, similar to trastuzumab and pertuzumab. In contrast, r40 is fully rabbit-derived and therefore requires a rabbit-specific secondary antibody for detection. Importantly, m66 showed binding intensity comparable to that of trastuzumab and pertuzumab. BT-474 and SK-BR-3 cells showed a significantly greater MFI than MCF7 cells, consistent with their higher HER2 expression levels (Fig 3A–3C).

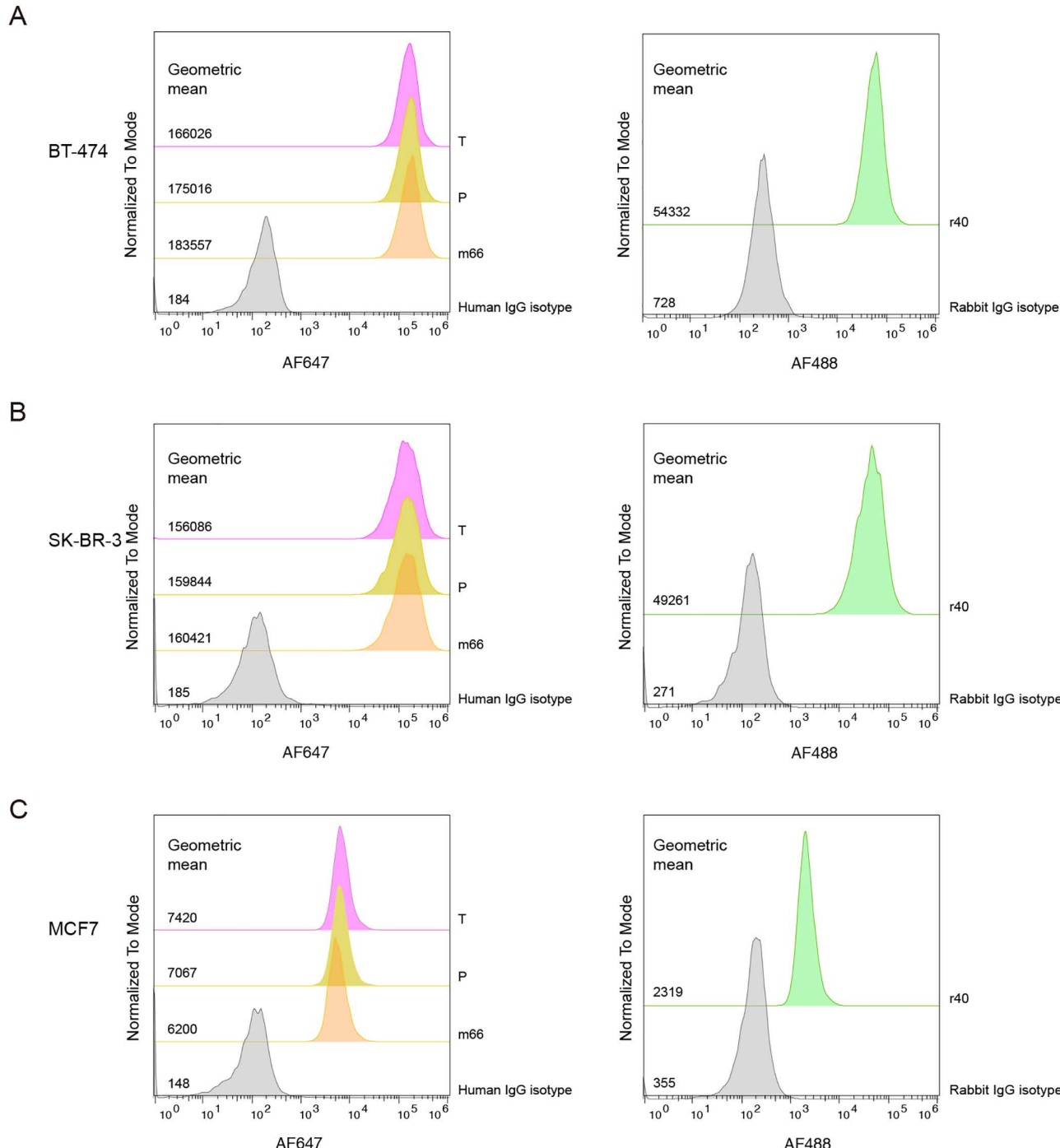

**Fig 3. Binding profiles of trastuzumab, pertuzumab, m66 and r40 to HER2-expressing breast cancer cell lines.** Antibody binding to (A) BT-474, (B) SK-BR-3 and **(C)** MCF7 was evaluated by flow cytometry. Cells were incubated with 267 nM of each antibody and detected using fluorescence-labeled secondary antibodies.

## Antibody r40 inhibits ligand-independent HER2 signaling pathways

The PI3K/AKT and MAPK pathways represent major signaling cascades downstream of the EGFR family [1,23]. Both trastuzumab and pertuzumab have been shown to inhibit these pathways in the breast cancer cells [12,29–31]. To evaluate the effects of our antibodies on the key signaling molecules, we treated cells with 10 µg/mL (66.7 nM) antibodies for 24 h and examined downstream signaling responses (Fig 4A). In MCF7 cells, while trastuzumab, r40, trastuzumab-pertuzumab combination, and the triple combination all significantly reduced HER2 protein levels, neither p-AKT (S473) nor p-ERK (T202/Y204) levels showed significant changes (Fig 4A). An increase in p-AKT (S473) and a decrease in p-ERK were also observed following pertuzumab treatment (Fig 4A). Consistent with the literatures, evaluating PI3K/AKT and MAPK pathways in MCF7 cells typically requires ligand stimulation (e.g., EGF or NRG1) [29,32,33]. Therefore, studying the impact of antibody drugs on signaling in MCF7 cells without EGF or NRG1 stimulation may have limited interpretative value.

In BT-474 cells, r40 antibody reduced p-AKT (S473) to a level comparable to that achieved by trastuzumab, and both antibodies exhibited greater potency than pertuzumab alone (Fig 4B). The trastuzumab-pertuzumab combination showed superior efficacy in reducing p-AKT (S473), and the triple antibody combination resulted in the lowest observed levels. In contrast, the m66 antibody had no effect on p-AKT (S473) levels. These changes in phosphorylation consistently correlated with corresponding HER2 protein expression levels (Fig 4B). Regarding the MAPK signaling pathway, among the four antibodies, only trastuzumab reduced p-ERK (T202/Y204) levels (Fig 4B). This reduction was not enhanced by the combination of trastuzumab and pertuzumab, and the triple antibody combination produced the greatest suppression of p-ERK (T202/Y204) (Fig 4B).

In SK-BR-3 cells, compared to trastuzumab, the r40 antibody reduced HER2 levels but did not demonstrate superior efficacy in lowering p-AKT (S473); its effect was similar to that of pertuzumab and m66 (Fig 4C). A slightly greater reduction in p-AKT (S473) was observed with the triple antibody combination than with the dual combination (Fig 4C). In contrast, no significant effect on p-ERK (T202/Y204) levels was observed with any single antibody or combinations (Fig 4C).

In conclusion, based on our results obtained in the absence of ligand activation, r40 demonstrated efficacy comparable to trastuzumab and pertuzumab in disrupting constitutive HER2 homodimerization and heterodimerization in BT-474 and SK-BR-3 cells. Furthermore, the addition of r40 to trastuzumab and pertuzumab enhanced their inhibitory effect.

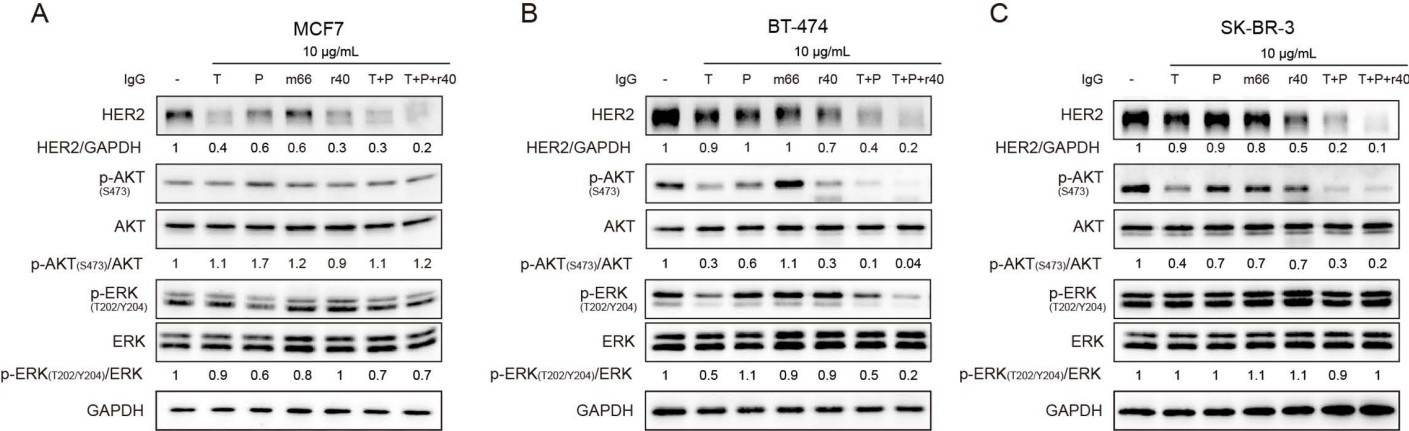

**Fig 4. Antibody r40 suppresses ligand-independent HER2 signaling in BT-474 and SK-BR-3 cells.** A-C. Immunoblots showing HER2 downstream signaling in (A) MCF7, (B) BT-474 and (C) SK-BR-3 cells treated for 24 h with 10 µg/mL of the indicated antibodies. Whole-cell lysates were immunoblotted with antibodies against the indicated proteins. For combination treatments, the indicated concentration refers to each antibody individually. The numerical values shown in the figures represent normalized quantitative data.

## Antibody r40 inhibits ligand-dependent HER2 signaling pathways

We next assessed the impact of antibody treatment on EGF-induced EGFR/HER2 heterodimerization and downstream signaling. EGF stimulation markedly increased phosphorylation of both AKT and ERK compared to the control (Fig 5A–5C).

In MCF7 cells, all the four antibodies reduced HER2 protein levels when administered individually (Fig 5A). Trastuzumab and r40 demonstrated a superior reduction in HER2 levels compared to pertuzumab and m66. The triple antibody combination led to a slightly greater reduction in HER2 levels than the dual combination (Fig 5A). Regarding p-AKT (S473), all treatments except m66 reduced its level, and the inhibitory effect of r40 was comparable to that of the trastuzumab and pertuzumab. In contrast, neither single antibodies nor combinations significantly affected p-ERK (T202/Y204) levels (Fig 5A).

In BT-474 cells, although neither r40 nor m66 reduced HER2 protein levels, r40 reduced both p-AKT (S473) and p-ERK (T202/Y204) and exhibited slightly greater inhibitory potency than trastuzumab or pertuzumab alone (Fig 5B). While m66 also decreased the phosphorylation of both kinases, its effect was less pronounced than that of r40. The triple antibody

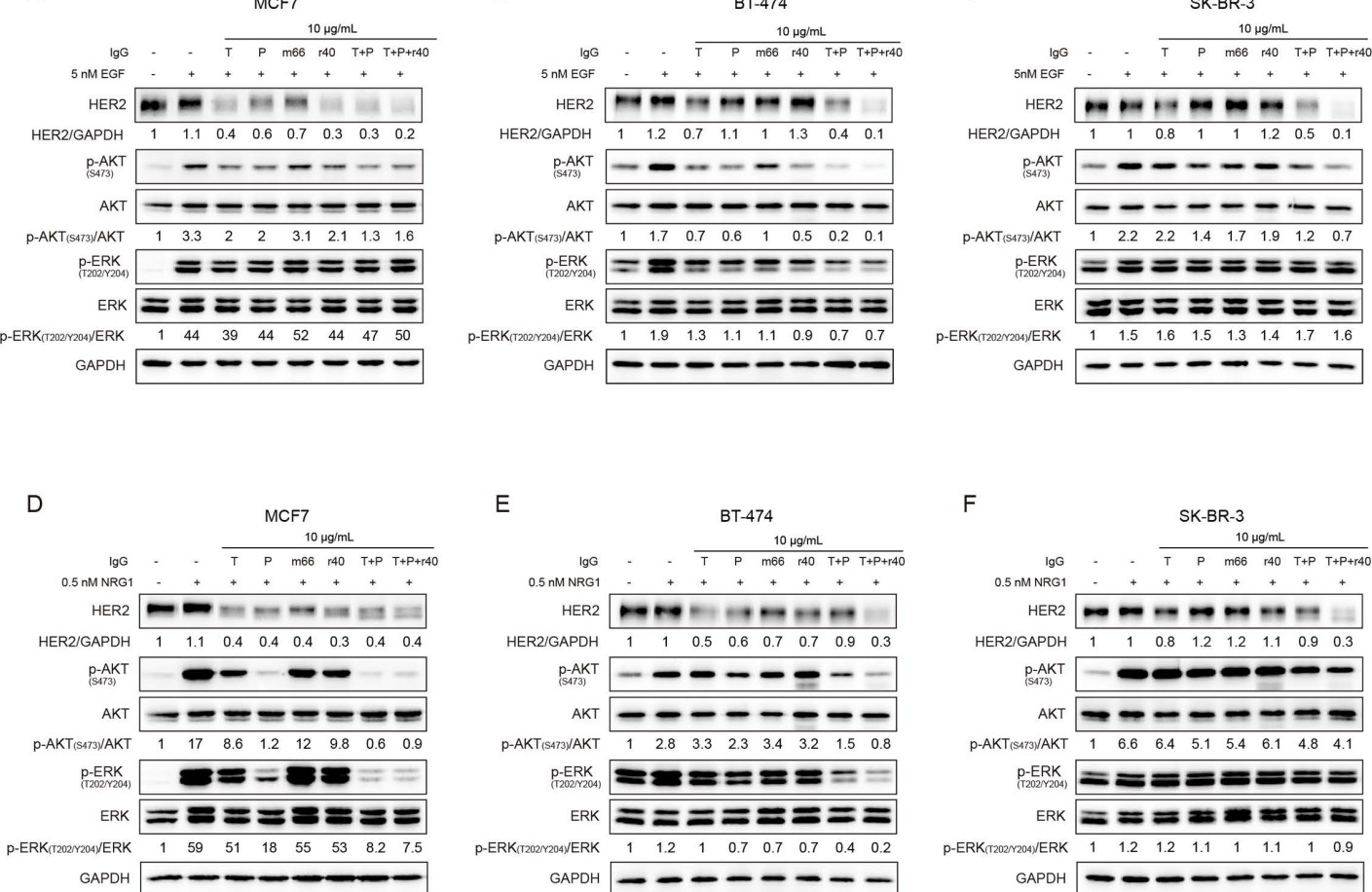

**Fig 5. Antibody r40 inhibits ligand-dependent HER2 signaling.** A-C. Immunoblots showing HER2 downstream signaling in (A) MCF7, (B) BT-474 and (C) SK-BR-3 treated with 10 µg/mL of the indicated antibodies for 24 h, followed by stimulation with 5 nM EGF for 10 min. D-F. Corresponding immunoblots of (D) MCF7, (E) BT-474 and (F) SK-BR-3 cells treated as above but stimulated with 0.5 nM NRG1 for 10 min. The numerical values shown in the figures represent normalized quantitative data.

combination produced a more pronounced reduction in p-AKT (S473) than the dual combination, whereas its suppressive effect on p-ERK (T202/Y204) was similar to that of the dual treatment (Fig 5B).

In SK-BR-3 cells, neither r40 nor m66 antibodies reduced HER2 protein levels (Fig 5C). When administered individually, both m66 and r40 demonstrated a modest reduction in p-AKT (S473), which was less pronounced than the effect achieved by pertuzumab. The triple antibody combination was more effective at reducing p-AKT (S473) than the dual antibody combination. Conversely, neither single agents nor antibody combinations significantly affected p-ERK (T202/Y204) levels (Fig 5C).

We next investigated the effect of antibody treatment on NRG1-induced HER2/HER3 heterodimerization and downstream signaling across the three breast cancer cell lines. In MCF7 cells, all four individual antibodies and their combinations significantly reduced HER2 protein levels (Fig 5D). Antibody r40 reduced the phosphorylation of AKT (S473) and ERK (T202/Y204) to a similar extent as trastuzumab compared to NRG1-stimulated control cells. Pertuzumab completely abrogated the NRG1-induced increase of both p-AKT (S473) and p-ERK (T202/Y204). Consequently, the dual and triple antibody combinations demonstrated comparable or even superior efficacy in suppressing these phosphorylation events (Fig 5D). In BT-474 and SK-BR-3 cells, both r40 and trastuzumab exerted limited effects on p-AKT (S473) and p-ERK (T202/Y204) (Fig 5E, 5F). In contrast, the triple antibody combination most strongly suppressed both p-AKT (S473) and HER2 levels in these cells (Fig 5E, 5F). Additionally, it significantly reduced the p-ERK (T202/Y204) levels in BT-474 cells (Fig 5E).

Taken together, r40 potently suppressed HER2-mediated signaling under both ligand-independent and ligand-dependent conditions and acted synergistically with trastuzumab and pertuzumab to enhance their antitumor efficacy.

## Complex assembly and structure determination

For structural characterization, full-length IgG antibodies were converted into the fragment antigen-binding (Fab) format (S1B, S1C Fig). To assemble the ternary complex of HER2, m66 Fab and trastuzumab Fab, we first isolated the HER2-m66 Fab binary complex using size exclusion chromatography (SEC). The SEC elution profile revealed complete complex formation between HER2 and m66 Fab, accompanied by a separate peak corresponding to the unbound m66 Fab (S3A Fig). Subsequent addition of trastuzumab Fab resulted in a clear forward shift of the HER2-m66 binary complex peak, confirming the formation of a stable HER2-m66-trastuzumab ternary complex (S3B–S3D Fig).

To assemble the tetrameric complex composed of HER2, r40, trastuzumab, and pertuzumab, we first purified the HER2-r40 binary complex, followed by the addition of trastuzumab Fab (S4A, S4B Fig). A clear shift in the SEC elution profile was observed upon subsequent addition of pertuzumab Fab, confirming the formation of the larger tetrameric complex (S4C–S4E Fig).

Single-particle cryo-EM reconstruction was used to determine the structures of the two HER2-antibodies complexes (S5 and S6 Figs). The cryo-EM density maps for the HER2-m66-trastuzumab and HER2-r40-trastuzumab-pertuzumab complexes were reconstructed and refined to overall resolutions of 3.2 Å and 3.1 Å, respectively. The final structural models were built with the HER2-trastuzumab-pertuzumab structure (PDB: 6OGE) [10] as a template and Alphafold 2 structure prediction. In the HER2-m66-trastuzumab complex, all three components were clearly resolved in the cryo-EM map, and a final high-resolution model was built and refined (Fig 6A). Trastuzumab binds to ECD IV, while m66 recognizes an epitope within ECD I of HER2. We performed a structural alignment between the HER2-trastuzumab-pertuzumab complex (PDB: 6OGE) and the HER2-m66-trastuzumab complex we determined with HER2 superimposed (Fig 6A). The binding site of pertuzumab is located at the dimerization interface of HER2. The observed inability of m66 and pertuzumab to bind HER2 simultaneously can be explained by a steric clash that would occur if both antibodies were bound. Superimposition of the HER2-HER3-NRG1 structure (PDB: 7MN5) [26] and the HER2-m66-trastuzumab structure demonstrates that m66 does not interfere with HER2 homodimerization and heterodimerization (Fig 6A).

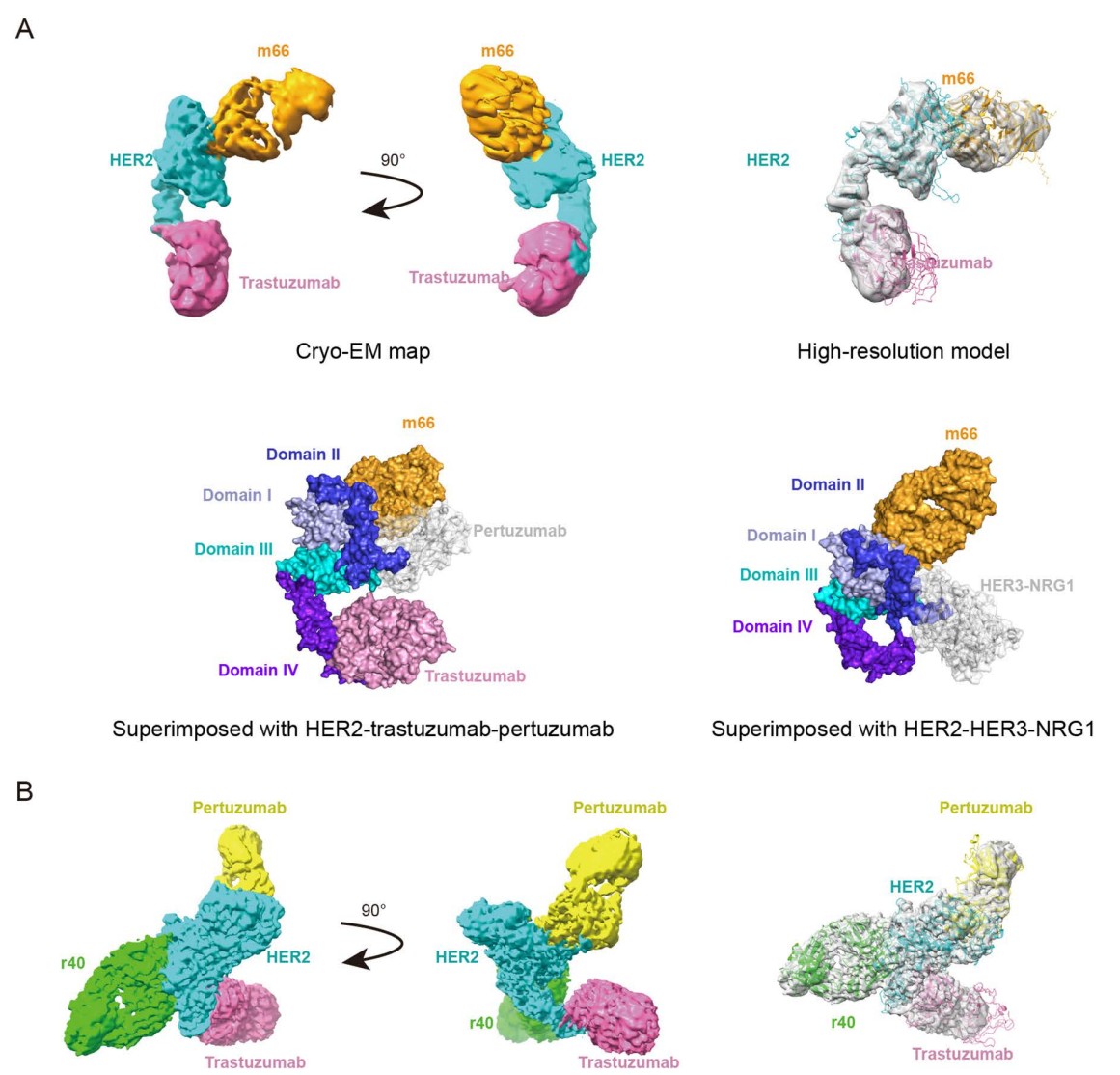

**Fig 6. Cryo-EM maps and structures of HER2-m66-trastuzumab and HER2-r40-trastuzumab-pertuzumab.** A. Cryo-EM map (upper left) depicting the complex composed of three components: HER2 (sky blue), m66 Fab (orange), and trastuzumab Fab (pink). The final model (upper right) incorporating the three components into the map. Superposition of HER2-trastuzumab-pertuzumab structure (PDB: 6OGE) and HER2-m66-trastuzumab structure

(lower left). Superposition of HER2-HER3-NRG1 structure (PDB: 7MN5) and HER2-m66-trastuzumab structure (lower right). B. Cryo-EM map (upper left) illustrated the tetrameric complex consisting of four components: HER2 (sky blue), r40 Fab (green), trastuzumab Fab (pink) and pertuzumab Fab (yellow). The final construction (upper right) with the four proteins fitted into the map. Superposition of HER2-trastuzumab-pertuzumab structure (PDB: 6OGE) and HER2-r40-trastuzumab-pertuzumab structure (lower left). Superposition of HER2-HER3-NRG1 structure (PDB: 7MN5) and HER2-r40-trastuzumab-pertuzumab complex (lower right).

In the tetrameric structure of HER2-r40-trastuzumab-pertuzumab, r40 binds to the ECD III/IV of HER2, a site topographically separate from the dimerization interface of HER2 (Fig 6B). The superimposed structure of HER2-HER3-NRG1 (PDB: 7MN5) and HER2-r40-trastuzumab-pertuzumab structure showed that the epitope recognized by antibody r40 lies opposite to the dimerization interfaces of HER2, indicating that r40 does not directly impede HER2 dimerization (Fig 6B). The extensive interaction interface between r40 and HER2 contributes to its high binding affinity and functional inhibition of HER2 signaling.

## Discussion

Antibody-based therapies have revolutionized cancer treatment by enabling highly specific targeting of tumor-associated antigens, thereby improving therapeutic outcomes and minimizing off-target effects. These agents constitute a cornerstone of contemporary oncology. Among approved anti-HER2 biologics, trastuzumab and pertuzumab recognize non-overlapping epitopes: trastuzumab binds to the juxtamembrane domain of ECD IV, whereas pertuzumab engages a region near the core of ECD II [10]. To date, no antibody therapeutics have been developed that specifically target ECD I or ECD III of HER2. Ongoing research continues to explore novel epitopes on HER2 to enable the development of more potent and complementary treatment strategies.

To obtain antibodies targeting epitopes distinct from those recognized by trastuzumab and pertuzumab, we employed ELISA and flow cytometry in the presence of saturating concentrations of both therapeutic antibodies. Our results indicate that m66 competes with pertuzumab for binding, while r40 recognizes an epitope that does not overlap with those of trastuzumab and pertuzumab. These findings were further corroborated by cryo-EM structural analyses, which confirmed the epitopes bound by m66 and r40.

In the absence of ligands, overexpressed HER2 can form homodimers [25]. Key ligands such as EGF and NRG1 play pivotal roles in the activation of HER2 signaling. EGF binding induces a conformational change in EGFR, transitioning it from a tethered to an extended conformation and exposing its dimerization arm. This structural shift enables interaction with the dimerization pocket of HER2, leading to the formation of a stable EGFR/HER2 heterodimer [34]. Similarly, NRG1—via its EGF-like domain— binds to HER3 and HER4, facilitating the formation of HER2/HER3 and HER2/HER4 heterodimers [26,28]. These heterodimers activate downstream PI3K/AKT and MAPK signaling pathways, thereby promoting cell proliferation. In this study, r40 potently suppressed HER2 signaling under both ligand-independent and ligand-dependent conditions, whereas m66 exerted a limited inhibitory effect.

Trastuzumab has been shown to facilitate the internalization of HER2, leading to its trafficking through endosomes and subsequent degradation in lysosomes [35]. In this study, under both ligand-dependent and ligand-independent conditions, the r40 antibody reduced HER2 protein levels in MCF7 cells, an effect comparable to that of trastuzumab (Figs 4A, 5A, 5D). In contrast, the m66 antibody—similar to pertuzumab, exhibited a comparatively weaker effect on HER2 expression. We propose that r40, similar to trastuzumab yet distinct from m66, induces the internalization of HER2. This difference may be attributed to the higher binding affinity of r40, and the formation of a stable antigen-antibody complex structure triggers the internalization process. In high HER2-expressing cell lines—BT-474 and SK-BR-3, individual antibodies were significantly less effective at reducing HER2 levels compared to their effects in low HER2-expressing MCF7 cells. However, the triple antibody combination exhibited the most potent reduction in HER2 expression across all cell types.

In summary, we screened a series of mouse- and rabbit-derived antibodies against human HER2 and identified r40 as an ECD III/IV-specific antibody that disrupted PI3K/AKT and MAPK signaling pathways and suppressed the growth of breast cancer cells. Notably, r40 enhanced the antitumor effect of the trastuzumab-pertuzumab combination. These findings position r40 as a promising therapeutic candidate capable of complementing existing HER2-targeted regimens, with potential clinical benefit for HER2-overexpressing cancers. Future studies should focus on elucidating its mechanism of action in greater detail and advancing the humanization of r40 for translational development.

## Materials and methods

### Cell culture

The human breast cancer cell lines BT-474, SK-BR-3 and MCF7 were obtained from the American Type Culture Collection (ATCC). These cells were maintained in DMEM (Gibco, 11995065) supplemented with 10% fetal bovine serum (NEW-ZERUM, FBS-UE500) and 100 μg/mL penicillin/streptomycin (Basalmedia, S110JV). Expi293 cells were cultured in Union 293 cell feed medium (Union, UP1000) containing 100 μg/mL penicillin/streptomycin.

### Plasmid construction and protein expression

The DNA sequence encoding human HER2 ECD (residues 23−652) was amplified and inserted into a modified pCAG vector containing a signal peptide and an N-terminal 9×His tag to enable the production of a secreted HER2 ECD fusion protein. A human rhinovirus (HRV) 3C protease cleavage site was incorporated between the His tag and HER2 ECD sequence (S1A Fig). Similarly, a synthetic DNA sequence encoding the 55 amino-acid EGF-like domain of human NRG1 (GTSHLVK-CAEKEKTFCVNGGECFMVKDLSNPSRYLCKCPNEFTGDRCQNYVMASF) was synthesized and cloned into the same modified pCAG vector. Both fusion proteins were expressed in Expi293 cells, purified from the culture medium using Chelating Sepharose™ Fast Flow (Cytiva, 17-0575-02), and subsequently treated with HRV 3C protease to remove the His tag.

### Plasmid construction and IgG expression

The variable heavy (VH) and light (VL) chains of trastuzumab, pertuzumab, m66, and m75 were cloned into the modified pCAG vectors containing the constant regions of human IgG1, with an HRV 3C protease cleavage site and a 9×His tag incorporated at the C-terminus of the heavy chain vector. The VH and VL regions of 39 rabbit antibodies were cloned into the similarly modified pCAG vectors carrying the rabbit IgG constant regions. Each IgG was obtained by co-transfecting of light chain and heavy chain plasmids into Expi293 cells. Following Ni-NTA affinity chromatography, His-tagged IgGs were directly eluted with imidazole, whereas non-His-tagged IgGs were released by HRV 3C protease cleavage following purification.

### ELISA

HER2 ECD protein was diluted to 0.2 ng/μL in PBS and used to coat the plates overnight at 4°C. After washing with PBST (PBS containing 0.05% Tween-20), the plates were blocked with blocking buffer (PBST supplemented with 3% BSA). Subsequently, the recombinant IgG-containing medium was diluted 1:100 in blocking buffer and added to the plates for a 1 h incubation. Following another wash, the bound IgG was detected using species-specific HRP-conjugated secondary antibodies: goat anti-human IgG1 Fc antibody (Millipore, AP113P) for trastuzumab, or goat anti-rabbit IgG (H+L) antibody (Abclonal, AS014) for rabbit monoclonal antibodies. After washing with PBST, TMB substrate (Beyotime, P0206) was added. The reaction was stopped by adding 1 M HCl, and the absorbance was measured at 450 nm using a microplate reader (Biotek).

### Pull-down

A 20 μL volume of a 20% slurry rProtein A/G magnetic beads (SMART, SM015010) was pre-equilibrated with washing buffer (PBS containing 0.1% CHAPS) and incubated with 10 μg of antibody at room temperature for 1 h. After three

washes with washing buffer, the beads were divided into two equal aliquots. One aliquot was used as the negative control; the other was incubated with 20 µg of HER2 ECD protein for 1 h at room temperature, washed three times with washing buffer, followed by SDS-PAGE analysis.

## Binding kinetics and affinity measurement

The binding kinetics between antibodies and HER2 were determined by SPRi using a PlexArray® system (Plexera). Antibodies in IgG format were immobilized at a concentration of 0.5 mg/mL on a three-dimensional optical crosslinking biosensor chip. A series of HER2 dilutions in PBS, ranging from 1 to 128 nM, underwent 285 s of association followed by 930 s of dissociation.

The $EC_{50}$ values were determined by ELISA. Briefly, ELISA plates (Corning, 3690) were coated with 0.2 ng/µL of HER2 solution overnight at 4 °C. After blocking, serial dilutions ($2^{-11}$-$2^{3}$ ng/µL) of antibodies were applied and incubated for 1 h at 37 °C. Subsequently, plates were incubated with either an HRP-conjugated goat anti-human IgG1 Fc antibody (Millipore, AP113P) or an HRP-conjugated goat anti-rabbit IgG (H+L) antibody (Abclonal, AS014) for 1 h at 37 °C. $EC_{50}$ values were calculated by fitting the binding curves in Graphpad Prism using a four-parameter logistic model (log (agonist) vs. response-variable slope).

## Epitope mapping of mouse antibodies by competitive ELISA

To determine the epitope binding regions of the two mouse antibodies, a competitive ELISA was performed. Briefly, ELISA plates were coated with 0.2 ng/µL HER2 ECD protein in PBS overnight at 4°C. After washing with PBST buffer, a non-His-tagged IgG was diluted to 2 ng/µL in blocking buffer and applied to the immobilized HER2 at 37°C for 1 h. Without removing the non-His-tagged IgG, a His-tagged antibody was then added directly to the mixture at 2 ng/µL and incubated at 37°C for 30 min with gentle shaking. Following a washing step, the binding of the His-tagged antibody was detected using an HRP-conjugated anti-His antibody (Genscript, A00612).

## Epitope mapping of rabbit antibodies using flow cytometry

To preliminarily characterize the epitopes of rabbit antibodies, a competitive binding assay was performed using flow cytometry. Expi293 cells were suspended in FACS buffer (DPBS containing 2% FBS) at a density of $1 \times 10^{7}$ cells/mL, then incubated with or without 10 µg/mL (66.7 nM) of trastuzumab and pertuzumab for 30 min. After washing, the cells were further incubated with 10 µg/mL of individual rabbit monoclonal antibodies for additional 30 min. Finally, the cells were stained with an AF488-conjugated goat anti-rabbit IgG secondary antibody (Invitrogen, A11008) and analyzed using a flow cytometer (Thermo Fisher Scientific) and the FlowJo software (version 10.8.1).

The saturation binding concentrations for trastuzumab and pertuzumab to HER2 on the cell surface of MCF7 were determined by flow cytometry. Briefly, MCF7 cells were resuspended in FACS buffer at $1 \times 10^{7}$ cells/mL and incubated with varying concentrations (0–400 nM) of trastuzumab and pertuzumab for 30 min. After incubation, the cells were stained with an AF647-conjugated goat anti-human secondary antibody (Invitrogen, A21445) and analyzed using a flow cytometer (Thermo Fisher Scientific). Data processing was performed using FlowJo 10.8.1 software.

The mouse antibodies (m66, m75) and rabbit-derived antibodies (r3, r4, r5, r29, r36, r40) were labeled with AF647 dye using NHS Ester chemistry. Briefly, each antibody was reacted with 20 molar equivalents of AF647 NHS ester (Biotium, 30–3007) for 2 h at room temperature protected from light. The reaction mixture was then dialyzed against PBS to remove the excess dye. For the competitive binding assay, serial 1:3 dilutions of the AF647-labeled antibodies (0–300 nM) were incubated with MCF7 cells that have been pre-saturated with 267 nM (twice the saturation concentration) of trastuzumab, pertuzumab, or a combination of both, as well as with untreated control cells.

## Tumor cell binding analysis by flow cytometry

BT-474, SK-BR-3 and MCF7 were briefly trypsinized at 37°C and the reaction was quenched with complete medium containing FBS. The cells were then collected by centrifugation. For each sample, $2 \times 10^5$ cells were resuspended in FACS buffer, and incubated with 267 nM (40 µg/mL) of either trastuzumab, pertuzumab, m66, r40 for 30 min at 4 °C. Human IgG isotype (Beyotime, A7001) and rabbit IgG isotype (Beyotime, A7016) served as negative controls. After incubation with the primary antibody, cells were stained with either an AF647-conjugated goat anti-human secondary antibody (Invitrogen, A21445) or an AF488-conjugated goat anti-rabbit secondary antibody (Invitrogen, A11008) for 30 min at 4 °C. Flow cytometry was performed to analyze antibody binding, and histogram overlays were generated to compare the staining intensity of the anti-HER2 antibodies with their corresponding isotype controls.

## Cell proliferation assay

BT-474, SK-BR-3, and MCF7 cells were seeded at a density of $1–2 \times 10^3$ cells per well in flat-bottomed 96-well cell culture plates and allowed to adhere overnight. The cells were then treated with a 1:3 serial dilution of each antibody (0–100 nM) for 4 d. Following the treatment period, cell viability was assessed using the Cell Counting Kit-8 (CCK-8, Selleck, B34302) according to the manufacturer's instructions. Absorbance was measured at 450 nm using a Synergy H1 microplate reader (BioTek).

## Immunoblotting

To evaluate ligand-independent HER2 signaling, MCF7, BT-474, and SK-BR-3 cells were treated for 24 h with 10 µg/mL trastuzumab, pertuzumab, m66, r40, the trastuzumab-pertuzumab combination, or the trastuzumab-pertuzumab-r40 triple combination. For ligand-dependent HER2 signaling, cells were first incubated with 10 µg/mL each antibody or combinations for 24 h, followed by stimulation with 0.5 nM recombinant human NRG1 or 5 nM EGF (Acro Biosystems, EGF-H52H3) for 10 min. After washing with ice-cold PBS, the cells were lysed, and the cell lysates were separated by 4–20% gradient SDS-PAGE under reducing conditions. Proteins were then transferred to a membrane and immunoblotted with the following primary antibodies: anti-GAPDH (Beyotime, AF0006), anti-AKT (Cell Signaling Technology, 9272), anti-phospho-AKT (Ser473) (Cell Signaling Technology, 4058), anti-p44/42 MAPK (Erk1/2) (Cell Signaling Technology, 4695), anti-phospho-p44/42 MAPK (Erk1/2) (Thr202/Tyr204) (Cell Signaling Technology, 4370), and anti-HER2 (Cell Signaling Technology, 4290). Western blot signals were quantified using Fiji software and normalized as follows: HER2 levels were normalized to GAPDH, p-AKT (S473) to total AKT, and p-ERK (T202/Y204) to total ERK.

## Fab purification and complex formation

The genes encoding the heavy and light chains of the trastuzumab, pertuzumab, m66, and r40 Fabs were assembled into a modified pCAG vector. A 9 × His tag was incorporated at the C-terminus of the heavy chain constant region to facilitate purification. Each Fab was obtained by co-transfecting of light chain and heavy chain plasmids into suspension Expi293 cells using polyethylenimine (PEI, Yeasen, 40816ES08). Following Ni-NTA purification, eluent was dialyzed to remove imidazole.

To assemble the HER2-m66-trastuzumab complex, HER2 was first incubated with an excess of m66 Fab. The mixture was then subjected to size-exclusion chromatography (SEC) using a Superdex 200 10/300 column (Cytiva) equilibrated with 20 mM HEPES pH 7.4, 150 mM NaCl. The HER2-m66 binary complex fractions were collected and subsequently incubated with an excess of trastuzumab Fab to form the ternary complex. The HER2-r40-trastuzumab-pertuzumab complex was prepared using a similar sequential assembly approach.

## Cryo-EM sample preparation and data collection

For cryo-EM grid preparation, 3 µL of HER2-m66-trastuzumab and HER2-r40-trastuzumab-pertuzumab (at 1.5 mg/mL) was applied to the freshly glow-discharged Quantifoil R1.2/1.3 Au grids. Samples were vitrified using a FEI Vitrobot Mark

IV (100% humidity, blot time 1 s). Subsequently, the grids were applied to a Titan Krios G3i or G4 transmission electron microscope (Thermo Fisher Scientific) with 300 kV accelerating voltage for data collection and characterization.

For the HER2-r40-trastuzumab-pertuzumab complex, 992 micrographs were automatically recorded on a Titan Krios G3i transmission electron microscope equipped with Gatan K3 direct electron detector, controlled by Serial-EM in the super-resolution counting mode. Data were collected at 81,000 × nominal magnification in the EFTEM mode, yielding a super-resolution pixel size of 0.532 Å on the image plane. The defocus range was set from −1.5 μm to −2.5 μm. The micrograph stack was dose-divided into 40 individual frames for dose-per-frame, with each frame receiving approximately 50 e-/ $Å^2$ and 2.3 s exposure time. The other images were captured automatically using a Titan Krios G4 transmission electron microscope equipped with a Falcon 4i direct electron detector. The micrographs were recorded in super-resolution counting mode using the EPU software at a nominal magnification of 130,000× in the NPTEM mode, resulting in a super-resolution pixel size of 0.466 Å on the image plane. With defocus values spanning between −1.5 μm to −2.5 μm, each micrograph stack was dose-fractionated into 1080 frames, receiving approximately 50 e-/ $Å^2$ total electron dose, under 3.51 s exposure time. For further processing, 2,439 and 2,965 micrographs were systematically acquired for the datasets of HER2-m66-trastuzumab and HER2-r40-trastuzumab-pertuzumab, respectively.

## Cryo-EM data processing

All cryo-EM datasets were processed with drift and beam-induced motion correction using MotionCor2 to perform whole-frame movie alignment [36]. The micrographs acquired on the Titan Krios G3i were subjected to a binning process with a factor of 2, yielding a calibrated pixel size of 1.064 Å/pixel, while those from the Titan Krios G4 datasets were similarly binned to a final pixel size of 0.932 Å/pixel. Defocus values were determined using Gctf [37] on dose-unweighted summed images. Subsequent cryo-EM data were carried out using RELION v3.1 [38,39] and cryoSPARC v2 [40], employing dose-weighted micrographs.

For the HER2-m66-trastuzumab dataset, a total of 2,017,757 particles were automatically selected through reference-free picking in RELION. These particles then underwent reference-free 2D classification in cryoSPARC, from which 466,601 particles representing well-defined classes were retained to generate an initial 3D model. Subsequently, 584,245 particles derived from high-quality 3D classes were subjected to heterogeneous refinement in cryoSPARC. The refined particle set was then loaded to RELION again for further rounds of 2D and 3D classification. From these, 282,644 particles corresponding to well-defined 3D classes were selected and subjected to heterogeneous refinement in cryoSPARC with CTF correction, ultimately yielding a final reconstruction of the HER2-m66-trastuzumab complex at 3.2 Å resolution.

For the HER2-r40-trastuzumab-pertuzumab dataset acquired using the Titan Krios G3i, a total of 1,490,027 particles were initially picked through an automatic particle selection process. These particles underwent reference-free 2D classification analysis in RELION to identify structural homogeneity. From the well-characterized 2D classes, 56,640 particles were chosen to advance to 3D classification. Ultimately, 24,229 particles from well-defined 3D classes were subjected to 3D auto-refinement, resulting in a final reconstruction of the HER2-r40-trastuzumab-pertuzumab complex at 9.1 Å resolution.

For the HER2-r40-trastuzumab-pertuzumab dataset acquired on the Titan Krios G4, an initial set of 2,293,703 particles was automatically selected using a reference derived from the previously reconstructed 9.1 Å resolution map. These particles underwent reference-assisted 2D classification in RELION. From the distinct 2D classes, 1,973,712 particles were selected and advanced to preliminary 3D classification analysis. A subset of 450,219 particles from high-quality 3D classes was subsequently utilized for 3D auto-refinement. Following Bayesian polishing, post-processing and CTF refinement, the final reconstruction reached a global resolution of 3.1 Å resolution reconstruction for the HER2-r40-trastuzumab-pertuzumab complex.

All observed resolutions were established based on the gold-standard Fourier shell correlation (FSC) criterion of 0.143 and the gold-standard FSC curves were validated to correct for soft mask effects using high-resolution noise substitution.

 

The cryo-EM maps were processed within the cryoSPARC software using module-based tools for B-factor optimization. Three-dimensional reconstructions were visualized and analyzed using UCSF Chimera or UCSF ChimeraX [41], and local resolution variations were determined via RELION analysis.

**Model building and structure refinement**

With the structure of HER2-trastuzumab-pertuzumab (PDB: 6OGE) [10] serving as the initial structural template, the cryo-EM density maps were utilized for model fitting. For the HER2-m66-trastuzumab complex, the HER2-trastuzumab module and the AlphaFold 2 [42] predicted structure of m66 were docked into the corresponding cryo-EM map as rigid-body fitting in UCSF Chimera [41]. Similarly, for the HER2-r40-trastuzumab-pertuzumab complex, the HER2-trastuzumab-pertuzumab structure (PDB: 6OGE) and the r40 structure predicted by AlphaFold 2 were incorporated into the cryo-EM density maps using UCSF Chimera for rigid-body fitting. All structural models were generated using COOT [43]. Maps and models in the figures were created using PyMOL (https://pymol.org/), UCSF Chimera, or UCSF ChimeraX. Detailed statistics for map reconstruction and model refinement are available in S1 Table.

**Supporting information**

**S1 Fig. Expression and purification of recombinant proteins and antibodies.** A. Schematic representation of the recombinant HER2 ECD and NGR1 EGF-like domain constructs. SP, signal peptide. B. Construct designs for the IgGs and Fabs used in this study. SP, signal peptide. C. SDS-PAGE analysis under both non-reducing and reducing conditions confirming the purity and integrity of the purified anti-HER2 IgGs and Fabs. D. Pull-down assay assessing the binding specificity of mouse antibodies. NC, negative control. E. Pull-down assay evaluating the reactivity of rabbit antibodies. F. ELISA for characterizing the affinity and specificity of rabbit monoclonal antibodies. G. Half-maximal effective concentration (EC$_{50}$) values of the indicated antibodies, as determined by ELISA.
(PNG)

**S2 Fig. Epitope binning of rabbit anti-HER2 antibodies by flow cytometry.** A. Binding of rabbit anti-HER2 antibodies (10 µg/mL) to Expi293 cells, compared to the rabbit isotype control IgG. B. Competitive binding profile of 36 rabbit antibodies on Expi293 cells, assessed by flow cytometry. The x-axis identifies each antibody clone; the y-axis shows the ratio of MFI in the presence versus absence of trastuzumab and pertuzumab (each at 10 µg/mL). C. EC$_{50}$ values of individual antibodies, the trastuzumab-pertuzumab combination, and the triple combination, determined by CCK-8 assay in three breast cancer cell lines.
(PNG)

**S3 Fig. Assembly and validation of the HER2-m66-trastuzumab ternary complex by SEC and SDS-PAGE.** A. SEC profile of the HER2-m66 binary complex, showing elution ahead of unbound m66 Fab. B. SEC trace after incubating the purified HER2-m66 binary complex with trastuzumab Fab, confirming the formation of a ternary complex. C. SDS-PAGE analysis under reducing conditions validates the composition and purity of HER2, HER2-m66 binary complex and HER2-m66-trastuzumab ternary complex. D. Schematic representation of the complex assembly process: HER2 ECD first binds m66 Fab to form a binary complex, which further incorporates trastuzumab Fab to generate a ternary complex.
(PNG)

**S4 Fig. Assembly of the HER2-r40-trastuzumab-pertuzumab complex monitored by SEC and SDS-PAGE.** A. SEC profile showing the formation of the HER2-r40 Fab binary complex. B. SEC trace after incubation of the purified HER2-r40 complex with trastuzumab Fab. C. SEC profile of the final tetrameric complex following further incubation with pertuzumab Fab. D. SDS-PAGE analysis under reducing conditions of the purified HER2 ECD, HER2-r40, HER2-r40-trastuzumab and

HER2-r40-trastuzumab-pertuzumab complexes. E. Schematic illustrating the sequential assembly: HER2 binds sequentially to r40 Fab, trastuzumab Fab, and pertuzumab Fab to form the tetrameric complex.
(PNG)

**S5 Fig. Cryo-EM data collection and image processing workflow for the HER2-m66-trastuzumab complex.** (A) Representative cryo-EM raw micrograph. (B) Results of 2D classification. (C) Schematic workflow of cryo-EM image processing and 3D reconstruction steps. (D) The upper panel displays the Fourier shell correlation (FSC) curve, while the lower left and right panels illustrate the distribution of cryo-EM map orientations and estimates of local resolution, respectively.
(PNG)

**S6 Fig. Cryo-EM data collection, image processing and reconstruction of the HER2-r40-trastuzumab-pertuzumab complex.** (A) Representative cryo-EM raw micrograph. (B) Results of 2D classification. (C) Schematic diagram for cryo-EM image processing and 3D reconstruction workflow. (D) The upper panel shows the FSC curve, while the lower left and right panels illustrate the distribution of cryo-EM map orientations and estimates of local resolution, respectively.
(PNG)

**S1 Table. Statistics of cryo-EM data collection, refinement and validation.**
(PNG)

**S1 Data. Raw data of ELISA, flow cytometry, CCK8, SPRi, Western blot, SEC.**
(XLSX)

**S1 Raw Images. Uncropped western blot and gel images.**
(PDF)

## Acknowledgments

We thank the Center of Cryo-Electron Microscopy, Core Facility of Shanghai Medical College, Fudan University for the supports on cryo-EM data collection and data analyses.

## Author contributions

**Investigation:** Chunchun Liu.

**Methodology:** Chunchun Liu, Xuan Luo, Peiyun Zhou, Xiangjuan Du, Xue Han, Ping Wang, Dan Zhao.

**Project administration:** Yanhui Xu, Dan Zhao, Huirong Yang.

**Software:** Yulei Ren, Qianmin Wang, Xue Han.

**Supervision:** Yanhui Xu, Huirong Yang.

**Visualization:** Yulei Ren, Qianmin Wang.

**Writing – original draft:** Chunchun Liu.

**Writing – review & editing:** Huirong Yang.

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
