## [Decision Letter · Decision Letter 0]

6 May 2025

Dear Dr. Liu,

The data presented in the manuscript need to be extended and updated to support the authors hypothesis (see the Editor's comments below)

We look forward to receiving your revised manuscript.

Kind regards,

Irina V. Lebedeva, Ph.D.

Academic Editor

PLOS ONE

Journal Requirements:

2.Thank you for stating the following financial disclosure: [Shanghai Natural Science Foundation of China (21ZR1407900)].

3. Thank you for stating the following in the Acknowledgments Section of your manuscript: [We thank the Center of Cryo-Electron Microscopy, Core Facility of Shanghai Medical College, Fudan University for the supports on cryo-EM data collection and data analyses. This work was supported by grants from the Shanghai Natural Science Foundation of China (21ZR1407900).]

6. PLOS requires an ORCID iD for the corresponding author in Editorial Manager on papers submitted after December 6th, 2016. Please ensure that you have an ORCID iD and that it is validated in Editorial Manager. To do this, go to ‘Update my Information’ (in the upper left-hand corner of the main menu), and click on the Fetch/Validate link next to the ORCID field. This will take you to the ORCID site and allow you to create a new iD or authenticate a pre-existing iD in Editorial Manager.

Additional Editor Comments:

The manuscript described generation and characterization of the novel monoclonal anti-HER-2 – specific antibodies. Two identified antibodies, murine m66 and rabbit r40, demonstrated inhibitory activity on growth of breast cancer cells and are shown to target different epitopes than clinically accepted trastuzumab and pertuzumab. Monoclonal rabbit r40 antibody could be developed further for testing in the clinical application.

The data presented in the manuscript are not quite sufficient to support the authors conclusions, and there are a few concerns that the authors need to address.

Antibody generation:What formats were the recombinant antibodies produced using the selected sequences from B cells?Protocols for ELISA and “pull-down” screening assays have to be provided.How did the authors confirmed selectivity and specificity of the antibodies?Competition assayWhat is “supersaturating IgG” [concentration] – how it was determined? What was the concentration of His-tagged IgG? There was no single dose-response binding curve for any reported antibody in the whole manuscript.Why mouse antibodies were tested via ELISA, and rabbit – via flow cytometry?Flow cytometry competitive experiment was not described adequately in the Results section. If the authors used transient transfection, how was the efficiency of the transfection tested? What was the concentrations for all used antibodies? How were these concentrations determined?Inhibition breast cancer cell proliferation assayThere was no figure that allows to compare the levels of HER2 expression in three cell line used. Binding dose-response curves with tested antibodies (including clinical references trastuzumab and pertuzumab) are required for each of the cell lines before the other experiments are performed. The data presented on Figure 3 are not sufficient do describe the tested antibodies binding abilities.What concentration of the tested antibodies did induce the inhibitory effect and what were the EC50 s for the inhibition? What criteria did the authors use to compare the inhibitory effect for all 4 antibodies? It looks that the effective concentrations for the tested antibodies were way too high (100µg/mL equal 67nM)Why the data presented in a different manner for different cells (Figure 2).Why NRG1-independent and NRG1-dependent proliferation experiment was performed only in MCF7 cells?HER2 signalingAgain, what was the rationale to use one or another cell line in different experiments? The authors either need a good rationale to use one cell liner, or use all three and show the results for all three.What was a rationale for using 50µg/mL of the antibodies? This is unusually high concentration for an antibody, especially for the antibodies with the reported high affinity – in this manuscript, the novel antibodies exhibit sub-nanomolar (r40) and low nanomolar (m66) affinity.Western blot images need quantification.Interestingly, r40 affects the AKT pathway in SK-BR-3 cell line but not in BT-474 (Figure 4). Is it a cell-line dependent effect, or there is a rational reason for it?For the ligand-dependent signaling, r40 affected AKT pathway in BT-474, and, to least extent, in SK-BR-3 but not in MCF7. Did the authors consider this difference?Finally, pharmacological inhibitors of the AKT and ERK are needed as controls for these type of the experiments

Reviewers' comments:

Reviewer's Responses to Questions

**Comments to the Author**

1. Is the manuscript technically sound, and do the data support the conclusions?

Reviewer #1: Yes

Reviewer #2: Yes

Reviewer #3: Yes

2. Has the statistical analysis been performed appropriately and rigorously?

Reviewer #1: Yes

Reviewer #2: Yes

Reviewer #3: Yes

3. Have the authors made all data underlying the findings in their manuscript fully available?

Reviewer #1: Yes

Reviewer #2: No

Reviewer #3: Yes

4. Is the manuscript presented in an intelligible fashion and written in standard English?

Reviewer #1: Yes

Reviewer #2: No

Reviewer #3: Yes

Reviewer #1: These authors sought to identify a novel antibody specific to HER2-expressing cancers that could combat treatment resistance and cancer recurrence. They found that r40 could inhibit several key pathways, including cell proliferation, by modulating several cellular pathways such as PI3K/AKT and MAPK.

I found this paper very interesting to read and I congratulate the authors on their novel work. My main comments are around sentence structure and grammatical/spelling errors. I found that the authors would often switch between past and present tense when discussing their results. For clarity and readability, it is important to pick one tense and stick to it throughout. I also found that the authors sometimes abbreviated terms with the acronym coming before the term. For example:

"The EGFR (human epidermal growth factor receptor) family comprises four members..."

However, this should read:

"The human epidermal growth factor receptor (EGFR) family comprises four members..."

Other times the authors correctly put the acronym in brackets afterwards. Please ensure this is consist across the manuscript. I have attached my revisions in a word doc. Again, I would like to congratulate the authors on a novel study that address an urgent gap in clinical care for patients with HER2 cancers.

Reviewer #2: This manuscript demonstrated the selection of mouse antibody m66 and rabbit antibody r40, which target distinct sites on HER2 that are different from those bound by trastuzumab and pertuzumab. It also examined their effects on inhibiting HER2-dependent signaling pathways.

Unfortunately, this manuscript does not include data from cancer cell proliferation inhibition assays using these antibodies in combination. As a result, the insights gained from the structural analysis of the antibody-HER2 complexes are not fully utilized, diminishing the impact of the report.

Additionally, while the experiments were conducted properly and yielded new findings, the results were presented in a manner that could lead to erroneous conclusions.

Consequently, it has been determined that, in its current form, this manuscript is insufficient for publication.

Throughout the manuscript, the initial citations of figures are not appropriately placed. Figures should be referenced within specific explanatory contexts. For example, Fig 1A should be cited in the sentence: "As expected, each non-His-tagged IgG significantly blocked...".

Similar citation issues appear throughout the manuscript.

There are numerous parts of the text that are insufficiently explained. The authors should seek feedback from colleagues by having them review the manuscript.

Below are some examples:

- In the Abstract and Introduction sections, both r40 and m66 should be mentioned, along with a clarification that they are monoclonal antibodies derived from rabbit and mouse, respectively.

- The sentence, "The binding kinetics of trastuzumab, pertuzumab…" should specify the binding target, possibly HER2.

- A reference for PDB entry #7MN5 should be provided.

- Figure legends should provide more specific details to allow the experiments to be understood without referring to the Materials and methods section.

This manuscript requires extensive proofreading, as it contains numerous spelling mistakes and typographical errors. Ensure that the co-authors thoroughly review the content.

For example, in the sentence: "The final structural models were built with the HER2-trastuzumab-...", the reference for PDB entry #60GE may be [Ref. 10] rather than [Ref. 25].

About the Figures:

Fig 2C: There is no statistically significant difference between r40 and trastuzumab in the NRG1 ligand treatment of MCF7 cells. Therefore, statements such as "Antibody r40 shows a more potent effect than trastuzumab..." should be avoided to prevent misinterpretation.

Fig 3: The horizontal axes should be aligned to facilitate vertical comparisons, as demonstrated in Fig 1B. Additionally, the peak intensity of r40 in BT-474 appears to be very low. What could be the possible explanation?

Figs 4 and 5: Blots should be digitally quantified and, if possible, represented as graphs. Of course, images of the blots should be incorporated into the graphs. Ideally, statistical analysis should also be conducted, and the discussion should be based on those results.

Fig 6: It would be better to adjust the display order of the figures. Following the sequence in the Results section, the results for m66 and r40 should be presented separately for each, in the order of trimer, tetrameric complex, and superimposed structure.

Reviewer #3: A nice in vitro study of selecting a new antibodies to HER2. Study is methodologically sound, data are well presented and manuscript as a whole is written in logical manner. Cryo EM part of study is especially commendable.

**Do you want your identity to be public for this peer review?** For information about this choice, including consent withdrawal, please see our Privacy Policy

Reviewer #1: No

Reviewer #2: No

Reviewer #3: **Yes:** Benjamin Benzon

---

## [Author Response · Author response to Decision Letter 1]

24 Jul 2025

Thanks for the reviewers’ and editors' constructive comments and suggestion, we have significantly improved the manuscript by performing critical experiments and refining the manuscript, resulting in a substantial enhancement in quality. The main point is: the p-AKT level and HER2 protein level is significantly decreased with triple combination (trastuzumab-pertuzumab-r40) use in breast cancer cells, which shows that r40 is a promising candidate to augment existing HER2-targeted therapies, potentially improving outcomes for HER2-overexpressing cancers. Future work should include detailed mechanistic analyses and humanization of r40.

---

## [Decision Letter · Decision Letter 1]

28 Aug 2025

Dear Dr. Liu,

Thank you for submitting your manuscript to PLOS ONE. After careful consideration, we feel that it has merit but does not fully meet PLOS ONE’s publication criteria as it currently stands. Therefore, we invite you to submit a revised version of the manuscript that addresses the points raised during the review process.

We look forward to receiving your revised manuscript.

Kind regards,

Irina V. Lebedeva, Ph.D.

Academic Editor

PLOS ONE

Journal Requirements:

Additional Editor Comments:

There are a few discrepancies between the dose response curves and tabulated EC50 values (Fig. 2 and 6). Please check the numbers, values etc to make sure there is a correspondence in the data.

expi293 cells do not express HER2 naturally, please clarify the experimental details and conclusions. regarding the saturation concentration experiments, ususally the EC50 binding values are considered to be  saturating.

The western blots images must be quantified and normalized to the levels of house keeping proteins (in particular, because the levels of the latters are not even).

It is not quite clear why the authors didn't see the dose response effect when tested HER2 signaling - it may indicate that the effect is not induced by the antibodies. The authors should re-test the dose response to cover the bigger concentration range (i.e. 1: 5 dilution, 6-8 points). Could it also be that the effect is time-dependent, and was not tested at the optimal time point? Also, why the levels of HER2 are so different for untreated MCF-7 (Fig.4)?

Figure 2 is confusing. What is plotted on the X axis - concentration of the antibody or the logarithm of concentration?

Reviewers' comments:

Reviewer's Responses to Questions

**Comments to the Author**

Reviewer #1: All comments have been addressed

Reviewer #2: (No Response)

2. Is the manuscript technically sound, and do the data support the conclusions?

Reviewer #1: Yes

Reviewer #2: Yes

3. Has the statistical analysis been performed appropriately and rigorously?

Reviewer #1: Yes

Reviewer #2: Yes

4. Have the authors made all data underlying the findings in their manuscript fully available?

Reviewer #1: Yes

Reviewer #2: Yes

5. Is the manuscript presented in an intelligible fashion and written in standard English?

Reviewer #1: Yes

Reviewer #2: No

Reviewer #1: All feedback from previous reviews was successfully implemented. I found some spelling/grammatical errors in this round, and have highlighted these (green) in the manuscript. Other than this, I found this work to be very interesting and well communicated, and believe it is ready for publication once these errors have been corrected. Congratulations to the authors

Reviewer #2: I acknowledge the addition of new data in the revised manuscript, which contributes to strengthening the overall conclusions.

However, the data presentation remains confusing and poorly structured, making the manuscript, in its current form, unsuitable for publication.

To improve clarity and impact, revisions are needed.

The following are examples of suggestions for improvement.

1. Antibody screening should be described in the Materials and Methods section.

- To Supplemental data: ELISA (no data provided?), pull-down assay (Response Letter Fig. 1), and FCM using Expi293 cells (Fig. S2A)

- To M&M: T&P saturation data (Fig. 1B) should be explained as the basis for determining T&P concentrations at 267 nM.

2. The Results section should describe the selection process in comparison to T&P.

- Data: mouse antibody ELISA (Fig. 1A), rabbit antibody FCM using Expi293 cells (Fig. S2B), and FCM using MCF7 cells (Fig. 1C–E*, S2C)

*The r35 data in Fig. 1E is not required

*Legends should be standardized to “non-saturated cells (or MCF7)” and “T&P-, T-, or P-saturated cells (or MCF7)”

3. Emphasize the results of the CCK-8 assay (Fig. 2A, B*) and HER2 signaling analysis (Figs. 4, 5**).

*Including EC50 values (Response Letter Table 1) would be helpful, although the graph curves do not appear to support reliable EC50 estimation.

**Consider placing the HER2 blot at the top of the figure panel

4. Summarize the characteristics of m66 and r40.

- To Results or Supplemental data: SPR (Fig. 2C), FCM (Fig. 3*)

*Cite the paper (doi: 10.1155/2012/486568, Response Letter Figure 1) to support the low HER2 expression in MCF-7 cells

*Include units and labels on the y-axis

5. In Fig. 6, divide the panels into A (m66 group) and B (r40 group), and label each panel with consistent headings such as:

"Cryo-EM map", "High-resolution model", "Superimposed with HER2-T-P“, and "Superimposed with HER2-HER3-NPG1".

6. The manuscript should be carefully reviewed to ensure internal consistency across all parts. Several inconsistencies remain between the revised main text and other sections such as the Materials and Methods and figure captions.

Examples:

- The ELISA EC50 data (Response Letter Fig. 1) are not presented, yet the Materials and Methods section still describes them

- The caption for Fig. 6 does not correspond to the actual figure content

- The statement "When cells were pre-incubated with trastuzumab and pertuzumab, the mean fluorescence intensity (MFI) of the rabbit monoclonal antibody remained comparable to control cells incubated with the rabbit antibody alone" lacks supporting data

- In Fig. 5C, the claim that "m66 inhibited p-AKT (S473) more potently than r40" is not clearly supported by the band intensity shown

- The term "AlphaFold2" is used in the Results section, whereas "AlphaFold" appears in the Materials and Methods section—this should be clarified

7. A thorough proofreading is required. Numerous typographical and grammatical errors are present throughout the manuscript. Please ensure that basic language editing is completed prior to submission.

Examples:

- "While antibody m75 blocked pertuzumab and pertuzumab blocked m75 (Fig 1A)." — This sentence is grammatically incomplete and unclear.

- "..., which explained why m66 and pertuzumab didn’t not bind HER2 simultaneously."

- "..., and then a of serial dilutions of HER2 (1-128 nM) in PBS ..."

- "In summary, we screened a panel of human HER2 antibodies and ..." — This sentence may give the misleading impression that the authors screened human antibodies.

To avoid confusion, it should be revised to: "a panel of (mouse and rabbit) antibodies targeting human HER2."

**Do you want your identity to be public for this peer review?** For information about this choice, including consent withdrawal, please see our Privacy Policy

Reviewer #1: No

Reviewer #2: No

---

## [Author Response · Author response to Decision Letter 2]

28 Sep 2025

Journal Requirements:

If the reviewer comments include a recommendation to cite specific previously published works, please review and evaluate these publications to determine whether they are relevant and should be cited. There is no requirement to cite these works unless the editor has indicated otherwise.  Additional Editor Comments:There are a few discrepancies between the dose response curves and tabulated EC50 values (Fig. 2 and 6). Please check the numbers, values etc to make sure there is a correspondence in the data.

Fig. 1. The ELISA EC50 values as presented in GraphPad Prism 9 software.

Fig. 2. Determination of EC₅₀ values for trastuzumab, pertuzumab, m66, m75, and r40 binding to immobilized HER2 ECD by ELISA.

Thank you for the kind reminder. We have checked the values in the GraphPad Prism 9 (Fig. 1). The antibody concentrations used were 2-11-23 ng/μL, with log values ranging from -11 to 3. We have also corrected the unit in corresponding table from "ng/mL" to "ng/μL" and “nM” (Fig. 2). Based on the ELISA analysis, the EC50 values for trastuzumab, pertuzumab, m66, m75 and r40 binding to immobilized HER2 ECD are as follows: 0.004837 ng/μL, 0.006954 ng/μL, 0.01359 ng/μL, 0.009973 ng/μL and 0.004146 ng/μL (Fig. 2).

Expi293 cells do not express HER2 naturally, please clarify the experimental details and conclusions. Regarding the saturation concentration experiments, ususally the EC50 binding values are considered to be saturating.

Fig.3. HER2 RNA levels in MCF7, HEK293, BT-474 and SK-BR-3 cells based on The Human Protein Atlas.

Thanks for the editor's comment. According to data from the Human Protein Atlas, HER2 RNA expression levels are comparable between MCF7 and HEK293 cell lines, with nTPM values of 47.1 and 36.4, respectively (Fig. 3). In contrast, BT-474 and SK-BR-3 cells show substantially higher HER2 expression, with nTPM values of 1979.6 and 2448.4, respectively (Fig. 3).

We initially cloned the full-length HER2 protein and expressed it in Expi293 cells to evaluate antibody binding. However, we observed that untransfected Expi293 cells also exhibited binding to the HER2 antibodies. Hence, we proceeded with subsequent experiments without transfection.

Fig.4. Flow cytometry analysis of rabbit anti-HER2 antibody binding compared to isotype control in Expi293 cells

As shown in Fig. 4, multiple rabbit monoclonal antibodies bound to untransfected Expi293 cells, with fluorescence intensities ranging from 103 to 104, compared to the rabbit IgG isotype control. Consistent with this, Fig. 3 of the revised manuscript shows that HER2-high-expressing BT-474 and SK-BR-3 cells exhibited fluorescence intensities around 105, while HER2-low-expressing MCF7 cells showed values between 103 and 104. These results demonstrate that Expi293 cells express low-levels of HER2 protein on the cell surface.

Although we detected low levels of HER2 protein expression on the surface of Expi293 cells via flow cytometry, this cell line is not a breast cancer model and is therefore considered less relevant in this field of research. Shall we omit the Expi293-based epitope validation data from the revised manuscript? Instead, we have focused on the results obtained from the breast cancer cell line MCF7 for epitope characterization.

The Western blots images must be quantified and normalized to the levels of house keeping proteins (in particular, because the levels of the latters are not even).

We appreciate the editor’s comment. In response, we have quantified the Western blot signals and normalized them as follows: HER2 levels were normalized to GAPDH, p-AKT (S473) to total AKT, and p-ERK (T202/Y204) to total ERK. The normalized quantitative data are now provided in the revised Fig. 4 and Fig. 5.

It is not quite clear why the authors didn't see the dose response effect when tested HER2 signaling - it may indicate that the effect is not induced by the antibodies. The authors should re-test the dose response to cover the bigger concentration range (i.e. 1: 5 dilution, 6-8 points). Could it also be that the effect is time-dependent, and was not tested at the optimal time point? Also, why the levels of HER2 are so different for untreated MCF-7 (Fig.4)?

Thanks for the editor's comment. The reduced HER2 signal in the control sample of the 5 μg/mL antibody treatment group in the original Fig. 4A is likely attributable to an edge effect. This occurred because 14 samples were loaded on a single SDS-PAGE gel, and the large membrane size combined with the sample’s position at the outer edge may have led to diminished transfer efficiency or uneven antibody binding.

In Fig. 4B of the publication (doi: 10.4161/mabs.28786), dose-response assays were performed in MCF7 cells using trastuzumab and pertuzumab across a broad concentration range (0.01 to 50 nM, serially diluted at 1:2 or 1:3 ratios) under 0.5 nM NRG1 stimulation for 10 min. As demonstrated, both antibodies at 50 nM (7.5 μg/mL) markedly reduced the phosphorylation levels of AKT and MAPK.

Fig. 4B of the paper (doi: 10.4161/mabs.28786)

The MCF7 cell line is classified as an HR+ (ER+/PR+), HER2-low breast cancer model. This indicates that its basal level of HER2 protein expression is relatively low, with minimal formation of HER2 homodimers or heterodimers (e.g., with EGFR or HER3). Consequently, the basal activity of the PI3K/AKT and MAPK signaling pathways in these cells is likely not primarily driven by HER2. Therefore, treatment with HER2 antibody alone is unlikely to produce significant inhibitory effects on these signaling pathways�as demonstrated by our experiment (Fig. 4A of the revised manuscript). The addition of EGF or NGR1 is intended to "create" a sensitive system enabling the evaluation of HER2 antibody efficacy (Fig. 5A, 5D of the revised manuscript).

Therefore, using MCF7 cells in the absence of ligand stimulation to test antibody concentration gradients was not ideal. Accordingly, in the revised Fig. 4A of the manuscript, we have retained only the Western blot data for the 10 μg/mL antibody concentration.

In previously published literatures investigating the effects of trastuzumab and pertuzumab treatment on signaling pathway alterations in breast cancer cell lines via Western blot, the antibody concentrations commonly used by researchers range from 7.5 μg/mL to 15 μg/mL, with a treatment duration of 24 hours in most cases (Fig. 5). Therefore, we selected a concentration of 10 μg/mL to treat the three breast cancer cell lines and harvested the samples after 24 hours.

Fig.5. Previously published literatures investigating the effects of trastuzumab and pertuzumab treatment on signaling pathway alterations in breast cancer cell lines via Western blot

Figure 2 is confusing. What is plotted on the X axis - concentration of the antibody or the logarithm of concentration?

Thanks for the editor's comment. We would like to confirm that the X-axis represents antibody concentration (μg/mL) with a logarithmic (log₁₀) scale, as shown in Fig. 6.

Fig.6. The Format Axes as presented in GraphPad Prism 9.

Reviewers' comments:Reviewer's Responses to Questions

Comments to the Author1. If the authors have adequately addressed your comments raised in a previous round of review and you feel that this manuscript is now acceptable for publication, you may indicate that here to bypass the “Comments to the Author” section, enter your conflict of interest statement in the “Confidential to Editor” section, and submit your "Accept" recommendation.

Reviewer #1: All comments have been addressed

Reviewer #2: (No Response)

2. Is the manuscript technically sound, and do the data support the conclusions?The manuscript must describe a technically sound piece of scientific research with data that supports the conclusions. Experiments must have been conducted rigorously, with appropriate controls, replication, and sample sizes. The conclusions must be drawn appropriately based on the data presented.

Reviewer #1: Yes

Reviewer #2: Yes

3. Has the statistical analysis been performed appropriately and rigorously?

Reviewer #1: Yes

Reviewer #2: Yes

4. Have the authors made all data underlying the findings in their manuscript fully available?The PLOS Data policy requires authors to make all data underlying the findings described in their manuscript fully available without restriction, with rare exception (please refer to the Data Availability Statement in the manuscript PDF file). The data should be provided as part of the manuscript or its supporting information, or deposited to a public repository. For example, in addition to summary statistics, the data points behind means, medians and variance measures should be available. If there are restrictions on publicly sharing data—e.g. participant privacy or use of data from a third party—those must be specified.

Reviewer #1: Yes

Reviewer #2: Yes

5. Is the manuscript presented in an intelligible fashion and written in standard English?PLOS ONE does not copyedit accepted manuscripts, so the language in submitted articles must be clear, correct, and unambiguous. Any typographical or grammatical errors should be corrected at revision, so please note any specific errors here.

Reviewer #1: Yes

Reviewer #2: No

6. Review Comments to the AuthorPlease use the space provided to explain your answers to the questions above. You may also include additional comments for the author, including concerns about dual publication, research ethics, or publication ethics. (Please upload your review as an attachment if it exceeds 20,000 characters)

Reviewer #1: All feedback from previous reviews was successfully implemented. I found some spelling/grammatical errors in this round, and have highlighted these (green) in the manuscript. Other than this, I found this work to be very interesting and well communicated, and believe it is ready for publication once these errors have been corrected. Congratulations to the authors

Reviewer #2: I acknowledge the addition of new data in the revised manuscript, which contributes to strengthening the overall conclusions.However, the data presentation remains confusing and poorly structured, making the manuscript, in its current form, unsuitable for publication.To improve clarity and impact, revisions are needed.The following are examples of suggestions for improvement.1. Antibody screening should be described in the Materials and Methods section.- To Supplemental data: ELISA (no data provided?), pull-down assay (Response Letter Fig. 1), and FCM using Expi293 cells (Fig. S2A)

We thank the reviewer for the valuable suggestion. In response, we have prepared the corresponding figures (Fig. 7 and Fig. 8) and detailed experimental protocols for both the ELISA and pull-down assays. Would it be appropriate to include descriptions of these assays in the Materials and Methods section? Additionally, would you recommend including these two figures in the Supplementary Materials?

ELISA

HER2 ECD protein was diluted to 0.2 µg/mL in PBS and coated overnight at 4°C. After blocking, the medium of recombinant expressed IgG was diluted 1:100 in blocking buffer (PBST containing 3% BSA) and incubated for 1 h. Following washing steps, bound IgG was detected using the HRP-conjugated goat anti-human IgG1 Fc antibody (Millipore, AP113P) for trastuzumab, or HRP-conjugated goat anti-rabbit IgG (H+L) antibody (Abclonal, AS014) for rabbit monoclonal antibodies.

Fig. 7. ELISA Characterization of rabbit monoclonal antibodies.

Pull-down

A 20 μL volume of a 20% slurry rProtein A/G magnetic beads (SMART, SM015010) was pre-equilibrated with washing buffer (PBS containing 0.1% CHAPS) and incubated with 10 μg of antibody at room temperature for 1 h. After three washes with washing buffer, the beads were divided into two equal aliquots. One aliquot was used as the negative control; the other was incubated with 20 μg of HER2 ECD protein for 1 h at room temperature, washed three times with washing buffer, followed by SDS-PAGE analysis.

Fig. 8. Characterization of rabbit monoclonal antibodies by pull-down assay.

- To M&M: T&P saturation data (Fig. 1B) should be explained as the basis for determining T&P concentrations at 267 nM.

We thank the reviewer for the valuable suggestion. We have added “As shown in the saturation curve, the saturating concentrations for both antibodies were determined to be approximately 133.3 nM (Fig. 1B)” at page 6 of the revised manuscript. And “As expected, the binding of AF647-conjugated trastuzumab and pertuzumab was abolished in MCF7 cells pre-saturated with 267 nM (twice the saturation concentration) of unlabeled trastuzumab and pertuzumab (Fig. 1C)" in the following paragraph.2. The Results section should describe the selection process in comparison to T&P.- Data: mouse antibody ELISA (Fig. 1A), rabbit antibody FCM using Expi293 cells (Fig. S2B), and FCM using MCF7 cells (Fig. 1C–E*, S2C)

We thank the reviewer for the valuable suggestion.

For mouse antibody ELISA (Fig. 1A), the selection process in comparison to T&P is described as follows, “Furthermore, pre-saturation with m66 or m75 did not impair trastuzumab binding, suggesting that both antibodies bind to epitopes distinct from that of trastuzumab. Interestingly, the binding of m66 was impaired by pertuzumab, whereas pertuzumab binding was not affected by m66, indicating that m66 has lower binding affinity than pertuzumab and may bind an overlapping or adjacent epitope (Fig. 1A). Antibody m75 reduced the binding of pertuzumab, and conversely, pertuzumab impaired the binding of m75 (Fig. 1A). These findings suggest that m66 and m75 recognize epitopes distinct from that of trastuzumab, yet likely overlap with the binding site of pertuzumab.”

For rabbit antibody FCM using Expi293 cells (Fig. S2B), the selection process in comparison to T&P is described as follows, “To characterize rabbit antibody epitopes through competitive binding, we pre-incubated Expi293 cells with both trastuzumab and pertuzumab at the concentration of 10 μg/mL (66.7 nM). The cells were then incubated with individual rabbit antibodies and analyzed by flow cytometry to evaluate simultaneous binding. For the antibodies that did bind, pre-incubation with trastuzumab and pertuzumab did not reduce the mean fluorescence intensity (MFI) compared to the control (cells incubated with the rabbit antibody alone). This indicated that the binding of these rabbit antibodies to HER2 did not compete with the epitopes targeted by trastuzumab and pertuzumab. Based on these results, we concluded that six rabbit antibodies (r3, r4, r5, r29, r36, and r40) might recognize epitopes distinct from those of trastuzumab and pertuzumab (S2B Fig).”

For FCM using MCF7 cells (Fig. 1C–D), the selection process in comparison to T&P is described as follows, “Subsequently, trastuzumab, pertuzumab, m66, and m75 were conjugated with NHS-AF647 dye. As expected, the binding of AF647-conjugated trastuzumab and pertuzumab was abolished in MCF7 cells pre-saturated with unlabeled trastuzumab and pertuzumab (Fig. 1C). Similarly, the binding of both m66-AF647 and m75-AF647 was blocked exclusively by pre-saturation with unlabeled pertuzumab, but only minimally affected by trastuzumab saturation (Fig. 1D). These results further support that m66 and m75 recognize epitopes overlapping with that of pertuzumab, consistent with the ELISA data (Fig. 1A). Notably, increased m66-AF647 binding was observed in cells pre-saturated with trastuzumab, suggesting that trastuzumab binding induces conformational changes in HER2 that enhance the accessibility or affinity for m66 (Fig. 1D).”

For FCM using MCF7 cells (Fig. 1E), the selection process in comparison to T&P is described as follows, “The epitope recognition profiles of r3, r4, r5, r29, r36, and r40 were subsequently characterized and compared with those of trastuzumab and pertuzumab using the low-HER2-expressing MCF7 cells (Fig. 1E). Pre-saturation of HER2 with trastuzumab and pertuzumab did not impede the subsequent binding of AF647-conjugated r3, r4, r5, r29, or r40, suggesting that their epitopes are distinct from those targeted by these two therapeutic antibodies (Fig. 1E). In contrast, r36 exhibited enhanced binding

---

## [Decision Letter · Decision Letter 2]

26 Oct 2025

Dear Dr. Liu,

Thank you for submitting your manuscript to PLOS ONE. After careful consideration, we feel that it has merit but does not fully meet PLOS ONE’s publication criteria as it currently stands. Therefore, we invite you to submit a revised version of the manuscript that addresses the points raised during the review process.

We look forward to receiving your revised manuscript.

Kind regards,

Irina V. Lebedeva, Ph.D.

Academic Editor

PLOS ONE

Journal Requirements:

Reviewers' comments:

Reviewer's Responses to Questions

**Comments to the Author**

Reviewer #1: All comments have been addressed

Reviewer #2: (No Response)

2. Is the manuscript technically sound, and do the data support the conclusions?

Reviewer #1: Yes

Reviewer #2: Yes

3. Has the statistical analysis been performed appropriately and rigorously?

Reviewer #1: Yes

Reviewer #2: Yes

4. Have the authors made all data underlying the findings in their manuscript fully available?

Reviewer #1: Yes

Reviewer #2: Yes

5. Is the manuscript presented in an intelligible fashion and written in standard English?

Reviewer #1: Yes

Reviewer #2: Yes

Reviewer #1: I am satisfied that all previous suggestions have been implemented into the manuscript and I have no further recommendations. I believe this manuscript is ready for acceptance. Congratulations to the authors on their interesting work

Reviewer #2: The revised manuscript is significantly more readable and easier to follow. I would like to express my appreciation for the authors' dedicated efforts and commend their hard work.

Below are five points of concern regarding the revision.

If these are addressed, I believe the manuscript will be suitable for publication.

1. The modification of the two mouse antibodies into the human IgG format is not clearly explained. Since the purification of HER2 is described in the Materials and Methods section, it would be beneficial to include a comparable description of the antibody preparation, including whether the antibodies are in human IgG format and whether they are His-tagged.

2. The citation of Figure 3 in the following sentence appears to be unnecessary:

"The m66 antibody, which incorporates variable regions on a human IgG1 backbone, can be detected using an antihuman secondary antibody, similar to trastuzumab and pertuzumab."

3. The methods used for quantification and normalization (as explained in this Response Letter) of the Western blot bands should be included in the Materials and Methods section. Additionally, the legends for Figures 4 and 5 should clarify that the numerical values shown in the figures represent normalized quantitative data.

4. For the description of ECD I, III, and the III/IV interface, appropriate references should be added in the following sentences:

- In Introduction: "...ECD I plays a key role in forming a binding pocket for the dimerization arm of other EGFR family members. ECD III, located centrally within the extracellular domain, serves as a critical structural bridge between ECD II and ECD IV and interacts directly with ECD I."

- In Discussion: "...the ECD III/IV interface, a region where structural rearrangements are more likely to induce HER2 internalization."

5. In the following sentence, the citation [41] should be placed immediately after the term "AlphaFold 2":

"...the HER2-trastuzumab module and the AlphaFold 2[41] predicted structure of m66..."

Response to the authors' inquiry:

>> Would it be appropriate to include descriptions of these assays in the Materials and Methods section? Additionally, would you recommend including these two figures in the Supplementary Materials?

There is no issue as long as the methods and results of the ELISA and pull-down assays are properly described in the supplementary materials.

However, it is essential to cite them explicitly in the main text, specifically in the following sentence:

"...and evaluated for binding affinity using enzyme-linked immunosorbent assay (ELISA) and pull-down assays."

Furthermore, descriptions of ELISA methods, including EC50 and competitive assays, should be harmonized throughout the manuscript.

>> However, as Expi293 cells are not widely recognized for cell surface HER2 protein expression and are not commonly used in breast cancer research, shall we remove the corresponding experimental data and figures regarding HER2 antibody epitope determination in Expi293 cells from the revised manuscript?

This dataset provides evidence of an antibody selection process targeting epitopes distinct from those of T&P, and therefore should not be removed. Although the suitability of using Expi293 cells may be debatable, the experimental conditions are clearly described in the revised manuscript, so no further changes are necessary.

**Do you want your identity to be public for this peer review?** For information about this choice, including consent withdrawal, please see our Privacy Policy

Reviewer #1: No

Reviewer #2: No

---

## [Author Response · Author response to Decision Letter 3]

3 Nov 2025

1. The modification of the two mouse antibodies into the human IgG format is not clearly explained. Since the purification of HER2 is described in the Materials and Methods section, it would be beneficial to include a comparable description of the antibody preparation, including whether the antibodies are in human IgG format and whether they are His-tagged.

Thank you for the kind reminder. The methodology for IgG plasmid construction and purification has been included in the Materials and Methods section of the revised manuscript as follows.

Plasmid construction and IgG expression

The variable heavy (VH) and light (VL) chains of trastuzumab, pertuzumab, m66, and m75 were cloned into the modified pCAG vectors containing the constant regions of human IgG1, with an HRV 3C protease cleavage site and a 9× His tag incorporated at the C-terminus of the heavy chain vector. The VH and VL regions of 39 rabbit antibodies were cloned into the similarly modified pCAG vectors carrying the rabbit IgG constant regions. Each IgG was obtained by co-transfecting of light chain and heavy chain plasmids into Expi293 cells. Following Ni-NTA affinity chromatography, His-tagged IgGs were directly eluted with imidazole, whereas non-His-tagged IgGs were released by HRV 3C protease cleavage following purification.

2. The citation of Figure 3 in the following sentence appears to be unnecessary:"The m66 antibody, which incorporates variable regions on a human IgG1 backbone, can be detected using an antihuman secondary antibody, similar to trastuzumab and pertuzumab."

We appreciate the reviewer’s comment. We have removed the citation of Figure 3 in the revised manuscript.

3. The methods used for quantification and normalization (as explained in this Response Letter) of the Western blot bands should be included in the Materials and Methods section. Additionally, the legends for Figures 4 and 5 should clarify that the numerical values shown in the figures represent normalized quantitative data.

Thank you for the kind reminder. We have added the methods used for quantification and normalization of the Western blot bands in the Immunoblotting part of the Materials and Methods section as follows. “Western blot signals were quantified using Fiji software and normalized as follows: HER2 levels were normalized to GAPDH, p-AKT (S473) to total AKT, and p-ERK (T202/Y204) to total ERK” in the revised manuscript.

And we have added this sentence “The numerical values shown in the figures represent normalized quantitative data” in the legends of Fig. 4 and Fig. 5.

4. For the description of ECD I, III, and the III/IV interface, appropriate references should be added in the following sentences:- In Introduction: "...ECD I plays a key role in forming a binding pocket for the dimerization arm of other EGFR family members. ECD III, located centrally within the extracellular domain, serves as a critical structural bridge between ECD II and ECD IV and interacts directly with ECD I."- In Discussion: "...the ECD III/IV interface, a region where structural rearrangements are more likely to induce HER2 internalization."

Thank you for the kind reminder. The citations [26-28] have been placed after the following sentence in Introduction.

“Although neither ECD I nor ECD III is directly involved in HER2 dimerization, ECD I plays a key role in forming a binding pocket for the dimerization arm of other EGFR family members; ECD III, located centrally within the extracellular domain, serves as a critical structural bridge between ECD II and ECD IV and interacts directly with ECD I[26-28].”

According to this paper (doi: 10.1016/j.ctrv.2024.102826), antibodies directed against HER2 ECD I and IV are both capable of mediating HER2 internalization. The greater potency of the r40 antibody in reducing HER2 levels is more likely due to its higher affinity compared to the lower affinity of the m66 antibody. For this reason, the sentence “Alternatively, while m66 binding to ECD I may not elicit significant conformational changes, the epitope recognized by r40 lies at the ECD III/IV interface, a region where structural rearrangements are more likely to induce HER2 internalization” In Discussion has been removed in the revised manuscript.

5. In the following sentence, the citation [41] should be placed immediately after the term "AlphaFold 2":"...the HER2-trastuzumab module and the AlphaFold 2[41] predicted structure of m66..."

Thank you for the kind reminder. The citation has been placed immediately after the term "AlphaFold 2" in the sentence.

Response to the authors' inquiry:

>> Would it be appropriate to include descriptions of these assays in the Materials and Methods section? Additionally, would you recommend including these two figures in the Supplementary Materials?There is no issue as long as the methods and results of the ELISA and pull-down assays are properly described in the supplementary materials.However, it is essential to cite them explicitly in the main text, specifically in the following sentence:"...and evaluated for binding affinity using enzyme-linked immunosorbent assay (ELISA) and pull-down assays."Furthermore, descriptions of ELISA methods, including EC50 and competitive assays, should be harmonized throughout the manuscript.

We appreciate the reviewer's valuable comment. To address this point, we have expanded the Materials and Methods section to include the experimental details for the ELISA and pull-down assays. The relevant figures have been moved to the supplementary information and are now presented as S1D-S1F Fig. Furthermore, the descriptions of ELISA methods, including EC50 and competitive assays, have been harmonized throughout the manuscript.

>> However, as Expi293 cells are not widely recognized for cell surface HER2 protein expression and are not commonly used in breast cancer research, shall we remove the corresponding experimental data and figures regarding HER2 antibody epitope determination in Expi293 cells from the revised manuscript?This dataset provides evidence of an antibody selection process targeting epitopes distinct from those of T&P, and therefore should not be removed. Although the suitability of using Expi293 cells may be debatable, the experimental conditions are clearly described in the revised manuscript, so no further changes are necessary.

OK, thank you very much.

---

## [Decision Letter · Decision Letter 3]

18 Nov 2025

Inhibition of HER2 signaling and breast cancer cell growth with a novel antibody targeting HER2 ECD III/IV

PONE-D-25-13465R3

Dear Dr. Liu,

We’re pleased to inform you that your manuscript has been judged scientifically suitable for publication and will be formally accepted for publication once it meets all outstanding technical requirements.

Kind regards,

Irina V. Lebedeva, Ph.D.

Academic Editor

PLOS ONE

Additional Editor Comments (optional):

Reviewers' comments:

Reviewer's Responses to Questions

**Comments to the Author**

Reviewer #2: All comments have been addressed

2. Is the manuscript technically sound, and do the data support the conclusions?

Reviewer #2: Yes

3. Has the statistical analysis been performed appropriately and rigorously?

Reviewer #2: Yes

4. Have the authors made all data underlying the findings in their manuscript fully available?

Reviewer #2: Yes

5. Is the manuscript presented in an intelligible fashion and written in standard English?

Reviewer #2: Yes

Reviewer #2: (No Response)

**Do you want your identity to be public for this peer review?** For information about this choice, including consent withdrawal, please see our Privacy Policy

Reviewer #2: No

---

## [Editor Report · Acceptance letter]

PONE-D-25-13465R3

PLOS One

Dear Dr. Liu,

I'm pleased to inform you that your manuscript has been deemed suitable for publication in PLOS One. Congratulations! Your manuscript is now being handed over to our production team.

Kind regards,

on behalf of

Dr. Irina V. Lebedeva

Academic Editor

PLOS One